# TwinFlow: Realizing One-step Generation on Large Models with Self-adversarial Flows

**Zhenglin Cheng**[1,2,3,4,*]  **Peng Sun**[1,3,4,*,†]  **Jianguo Li**[3]  **Tao Lin**[1,3,‡]

[1]Westlake University  [2]Shanghai Innovation Institute  [3]Inclusion AI  [4]Zhejiang University

## Abstract

Recent advances in large multi-modal generative models have demonstrated impressive capabilities in multi-modal generation, including image and video generation. These models are typically built upon multi-step frameworks like diffusion and flow matching, which inherently limits their inference efficiency (requiring 40-100 Number of Function Evaluations (NFEs)). While various few-step methods aim to accelerate the inference, existing solutions have clear limitations. Prominent distillation-based methods, such as progressive and consistency distillation, either require an iterative distillation procedure or show significant degradation at very few steps (< 4-NFE). Meanwhile, integrating adversarial training into distillation (e.g., DMD/DMD2 and SANA-Sprint) to enhance performance introduces training instability, added complexity, and high GPU memory overhead due to the auxiliary trained models. To this end, we propose TwinFlow, a simple yet effective framework for training 1-step generative models that bypasses the need of fixed pretrained teacher models and avoids standard adversarial networks during training, making it ideal for building large-scale, efficient models. On text-to-image tasks, our method achieves a GenEval score of 0.83 in 1-NFE, outperforming strong baselines like SANA-Sprint (a GAN loss-based framework) and RCGM (a consistency-based framework). **Notably, we demonstrate the scalability of TwinFlow by full-parameter training on Qwen-Image-20B and transform it into an efficient few-step generator.** With just 1-NFE, our approach matches the performance of the original 100-NFE model on both the GenEval and DPG-Bench benchmarks, reducing computational cost by $100\times$ with minor quality degradation. Project page is available here.

Code: https://github.com/inclusionAI/TwinFlow

Models: https://huggingface.co/collections/inclusionAI/twinflow

## 1 Introduction

Modern generative paradigms—including diffusion (Ho et al., 2020; Song et al., 2020a), flow matching (Lipman et al., 2022; Ma et al., 2024), and consistency models (Song et al., 2023; Lu & Song, 2024)—have achieved state-of-the-art performance, forming the backbone of leading image and video generation systems (Peebles & Xie, 2023; Ho et al., 2022; Chen et al., 2024c; Xie et al., 2024a). Despite their success, these methods share a critical drawback: both training and sampling demand substantial computational resources. This challenge is magnified in the era of large-scale models (ModelTC, 2025). For these systems, efficient sampling is paramount, as the continuous, long-term cost of inference often surpasses the one-time training cost, directly impacting their economic viability and practical deployment (ModelTC, 2025; Xie et al., 2025a).

Numerous research efforts aim to accelerate generative inference by reducing the number of sampling steps. Early single-step methods, like Generative Adversarial Networks (GANs) (Goodfellow et al., 2014), often suffered from unstable training dynamics. To accelerate the multi-step diffusion models,

---

*Equal contributions. Work was done during internship at Inclusion AI, Ant Group.

†Project leader.

‡Corresponding author.

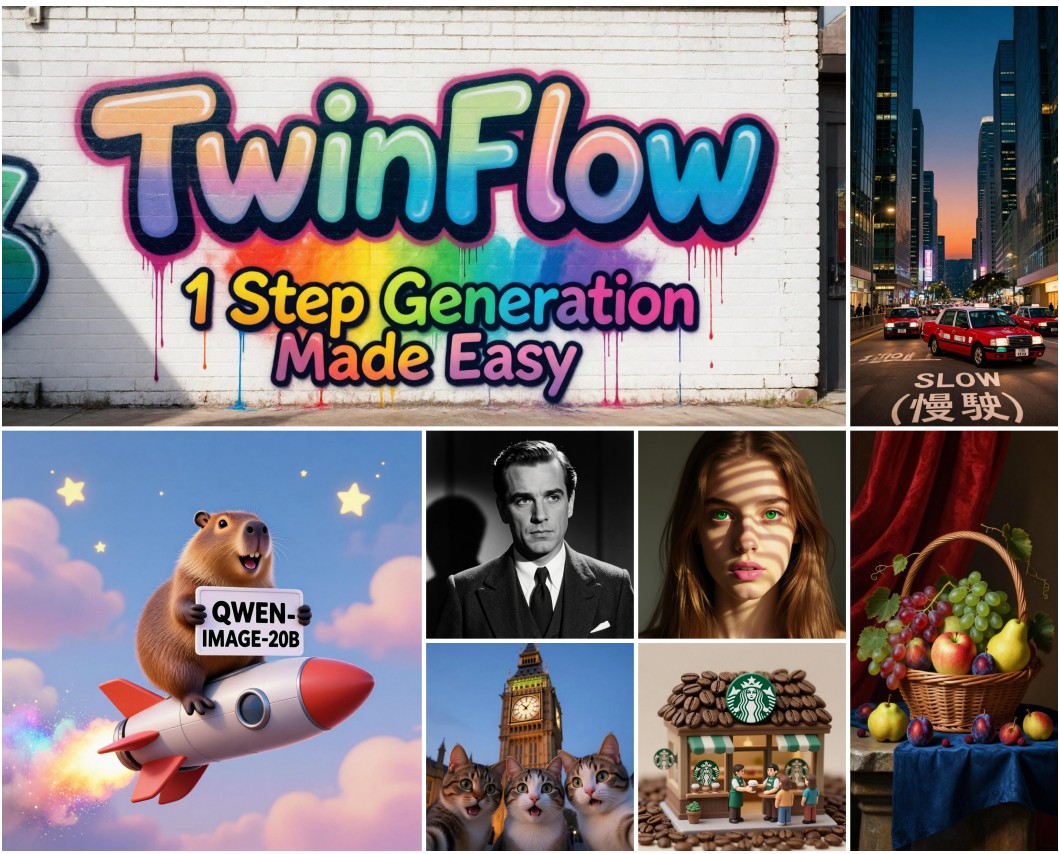

Figure 1: **Results of Qwen-Image-20B-TWINFLOW (NFE=2).** See prompts in App. E.3.

various distillation techniques have been introduced. These range from student-teacher methods, where a compact model learns to emulate a larger one in fewer steps (Salimans & Ho, 2022; Meng et al., 2022), to distribution matching distillation (e.g., DMD variants (Yin et al., 2024b;a)), which uses adversarial training to directly align the model's output distribution with the real data. In parallel, a powerful new paradigm of consistency models (Song et al., 2023) and their variants, such as LCM (Luo et al., 2023) and PCM (Wang et al., 2024a), has emerged, explicitly designed for high-quality generation in very few steps.

Despite their progress, existing few-step methods face a difficult trade-off between *simplicity*, *efficiency*, and *quality*. **(a)** *Complexity and instability:* As summarized in Tab. 1, adversarial methods such as GANs and DMD require auxiliary networks (e.g., discriminators) or frozen teacher models. This not only introduces training instability and sensitivity to hyperparameters but also increases architectural complexity and memory overhead (c.f. Fig. 2b), hindering their scalability to large models. **(b)** *Performance degradation:* Conversely, methods that train from scratch without adversarial guidance (Luo et al., 2023; Yin et al., 2024a), such as consistency models, often exhibit a sharp decline in quality at very low NFEs (< 4) (Chen et al., 2025c). *In summary*, we posit that existing methods either suffer from training instability, or require additional/frozen models (see our Tab. 1), which limits their simplicity and scalability in training large models.

To address these challenges, we propose TWINFLOW, a simple yet effective one-step generative training framework built on a novel twin-trajectory concept. By extending the standard time interval from $t \in [0, 1]$ to $t \in [-1, 1]$, we conceptualize two trajectories originating from the noise distribution. The positive branch ($t > 0$) maps noise to real data while the negative branch ($t < 0$) maps the same noise to "fake" data, enabling simultaneous learning of both transformations. Our learning objective is to minimize the discrepancy between the velocity fields of these two trajectories (see Fig. 2a). This forces the model to learn a more robust and direct mapping from noise to data, thereby enhancing 1-step generation performance in a self-supervised manner. As highlighted in Tab. 1, a key advantage of TWINFLOW is its simplicity, as it requires no auxiliary trained networks or frozen teacher models. Extensive experiments at different scales demonstrate the effectiveness of TWINFLOW, including

Table 1: **Comparison of different few-step generative modeling methods on their minimal dependence of auxiliary trained model and frozen teacher model.** Prior 1-step/few-step methods such as GAN requires a trained discriminator, diffusion/consistency distillation[*] requires a frozen teacher model, DMD requires training an auxiliary score function for fake data and a frozen teacher model, DMD2[†] trains a GAN discriminator and a fake score function at the same time. Our TWINFLOW achieves 1-step generation *without* depending on auxiliary trained or frozen models, offering high simplicity.

| Method | Generation type | #Auxiliary trained model | #Frozen teacher model |
|---|---|---|---|
| GAN (Goodfellow et al., 2014) | 1-step | 1 | 0 |
| Diffusion models (Ho et al., 2020) | multi-step | 0 | 0 |
| Flow matching models (Lipman et al., 2022) | multi-step | 0 | 0 |
| Diffusion distillation (Salimans & Ho, 2022) | few-step | 0 | 1 |
| Consistency training & distillation (Song et al., 2023) | 1-step, few-step | 0 | 0,1[*] |
| Distribution matching distillation (Yin et al., 2024b;a) | 1-step, few-step | 1,2[†] | 1 |
| **TWINFLOW (Ours)** | 1-step, few-step | 0 | 0 |

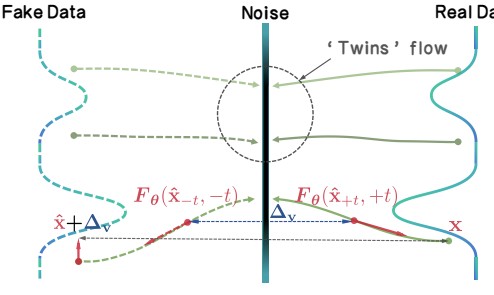

(a) **TWINFLOW overview.** The standard flow is on the right side (solid lines), its twin (dashed lines) is on the left side. The core of our method is to minimize of the difference between the velocity fields ($\Delta_\mathbf{v}$) of the standard flow and its twin flow.

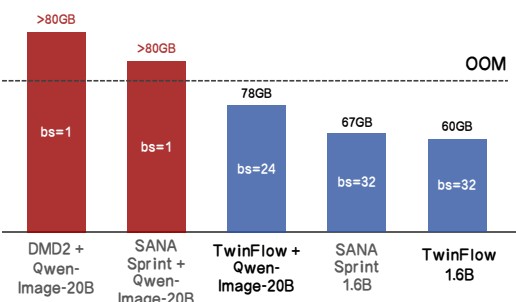

(b) **GPU memory comparison.** Directly adopting DMD2 and SANA-Sprint suffers from OOM when applying to ultra-large models. Our method can be easily applied to train Qwen-Image-20B.

Figure 2: **Overview of our TWINFLOW and training GPU memory comparison.** The GPU memory usage is measured on 1024×1024 resolution on Qwen-Image-20B (LoRA tuning) and SANA-1.6B.

text-to-image generation (cf. Sec. 4.2 & Sec. 4.3) on large models like Qwen-Image-20B (Wu et al., 2025a). 2-NFE visualizations on Qwen-Image-20B are given in Fig. 1. **Our key contributions are:**

(a) **Simple yet effective 1-step generation framework.** We propose a one-step generation framework that does not need auxiliary trained models (GAN discriminators) or frozen teacher models (different/consistency distillation), thereby eliminating GPU memory cost, allowing for more flexible and scalable training on large models.

(b) **Strong 1-NFE performance on text-to-image task.** Built on the any-step framework, TWIN-FLOW achieves strong text-to-image performance with only 1-NFE, achieving 0.83 GenEval score, surpassing SANA-Sprint (0.72) and RCGM (0.80).

(c) **Effective application on large models.** By applying TWINFLOW, we successfully bring 1/2-NFE generation capabilities to Qwen-Image-20B. We achieve GenEval score of 0.86 and DPG score of 86.52 (1-NFE); GenEval 0.87 and DPG Score 87.64 (2-NFE), which are highly competitive with the original 100-NFE scores of 0.87 and 88.32.

## 2 PRELIMINARIES

Given a dataset $\mathcal{D}$, let $p(\mathbf{x})$ represent its data distribution and $p(\mathbf{x}|\mathbf{c})$ the conditional distribution given a condition $\mathbf{c}$. Generative models aim to learn a transformation from a simple source distribution, $p(\mathbf{z})$, such as the standard Gaussian distribution $\mathcal{N}(\mathbf{0}, \mathbf{I})$, to the complex target distribution, $p(\mathbf{x})$.

**Any-step generative model framework.** A recent framework, RCGM (Sun & Lin, 2025), introduces a unified formulation for the any-step generation framework, covering paradigms like multi-step generative models (Ho et al., 2020; Song et al., 2020b; Lipman et al., 2022) and few-step generative models (Song et al., 2023; Lu & Song, 2024; Frans et al., 2024; Geng et al., 2025; Sun et al., 2025).

In this framework, a prediction function can be generally defined as $\boldsymbol{f}(\mathbf{x}_t, r) := \mathbf{x}_r - \mathbf{x}_t$, which predicts the target point $\mathbf{x}_r$ from the current point $\mathbf{x}_t$ along a specific PF-ODE trajectory. The unified training objective for any-step models is given by:

$$\mathcal{L}(\boldsymbol{\theta})_{\text{base}} = \mathbb{E}_{\mathbf{x}, \mathbf{z}, \{t_i\}_{i=0}^{N+1}} \left[ d \left( \frac{\mathrm{d}\mathbf{x}_t}{\mathrm{d}t}, \frac{1}{\Delta t} \left[ \boldsymbol{f}_{\boldsymbol{\theta}}(\mathbf{x}_t, t_{N+1}) - \sum_{i=1}^{N} \boldsymbol{f}_{\boldsymbol{\theta}^-}(\mathbf{x}_{t_i}, t_{i+1}) \right] \right) \right], \qquad (1)$$

where $\mathbf{x}_t = \alpha(t)\mathbf{z} + \gamma(t)\mathbf{x}$, $\mathbf{z} \sim \mathcal{N}(\mathbf{0}, \mathbf{I})$, $t \sim \mathrm{U}(0, T)$, $t_i \sim \mathrm{U}(t_{i-1}, 0)$, and $d(\cdot, \cdot)$ is a metric function. Under flow matching objective and linear transport, we have $\boldsymbol{f}_{\boldsymbol{\theta}}(\mathbf{x}_t, r) = \boldsymbol{F}_{\boldsymbol{\theta}}(\mathbf{x}_t, r) \cdot (t - r)$, where $\boldsymbol{F}_{\boldsymbol{\theta}}$ is a neural network, $\boldsymbol{F}_{\boldsymbol{\theta}^-}$ is the no grad version. In practice, we use `Network(x_t, t, r)` as the implementation of $\boldsymbol{F}_{\boldsymbol{\theta}}(\mathbf{x}_t, r)$. Equation (1) demonstrates how both multi-step and few-step frameworks can be seen as specific instances of the broader any-step framework, which we will detail below.

**Multi-step generative models.** Diffusion (Ho et al., 2020; Song et al., 2020b) and flow matching models (Lipman et al., 2022) can be derived from the RCGM framework. By setting $N = 0$, the objective in (1) reduces to their respective training objectives, where the predict function becomes $\boldsymbol{f}(\mathbf{x}_t, t - \Delta t)$ in the limit $\Delta t \to 0$. During sampling, these models iteratively solve the PF-ODE by integrating the velocity field $\frac{\mathrm{d}\mathbf{x}_t}{\mathrm{d}t}$, starting from a noise sample $\mathbf{x}_1 \sim p(\mathbf{z})$ at $t = 1$ and ending at $t = 0$ to obtain samples from $p(\mathbf{x})$.

**Few-step generative models.** Few-step models are also instances of the RCGM framework, typically corresponding to $N = 1$ case: (1) setting $t_1 = t - \Delta t, \Delta t \to 0$ and $t_2 = 0$ recovers the objective for consistency models (Song et al., 2023; Lu & Song, 2024); (2) setting $t_1 \in (t_2, t)$ corresponds to shortcut models (Frans et al., 2024), where the predict function is defined as $\boldsymbol{f}(\mathbf{x}_t, r) \leftarrow \boldsymbol{f}(\mathbf{x}_t, s) + \boldsymbol{f}(\mathbf{x}_r, s)$ and $r = t_2 \in [0, t]$; (3) setting $t_1 = t - \Delta t$ (with $\Delta t \to 0$) yields the MeanFlow objective (Geng et al., 2025).

In summary, the RCGM framework offers a unified perspective that integrates both multi-step and few-step paradigms, facilitating their analysis and application. See more related work discussion in Sec. B.

## 3 METHODOLOGY

Current few-step methods within the any-step framework (Sec. 2) struggle to achieve high-quality one-step generation without resorting to a GAN loss, which adds significant complexity. To solve this, we propose TWINFLOW, a simple and self-contained approach that enhances one-step performance directly within the any-step flow matching framework. Our key idea is the introduction of twin trajectories, which create an internal self-adversarial signal and thus eliminate the need for an external GAN loss (Sec. 3.1). The method works by minimizing a difference between a "fake" and a "real" velocity field, which should ideally be zero (Sec. 3.2). We conclude by demonstrating how to integrate TWINFLOW into the broader any-step framework and provide practical designs in Sec. 3.3.

### 3.1 TWIN TRAJECTORY FOR SELF-ADVERSARIAL TRAINING

A key innovation of our method is the introduction of twin trajectories, which feature time-steps symmetric around $t = 0$ (see Fig. 2a). This structure creates a self-contained, discriminator-free adversarial objective designed to directly enhance one-step generation performance.

**Creating self-adversarial objective.** The standard learning process operates on a time interval $t \in [0, 1]$: real data $\mathbf{x}$ is perturbed by $\mathbf{x}_t^{\text{real}} = \alpha(t)\mathbf{z} + \gamma(t)\mathbf{x}$, where $\mathbf{z} \sim \mathcal{N}(\mathbf{0}, \mathbf{I})$, $t \sim \mathrm{U}(0, 1)$. To create our self-adversarial objective (as well as the twin trajectories), we extend this time interval from $t \in [0, 1]$ to $t \in [-1, 1]$. The negative half of this interval, $t \in [-1, 0]$, is designated for learning a generative path from noise to "fake" data produced by the model itself.

Specifically, we task the network to learn the generative path to its own outputs. We take a fake sample $\mathbf{x}^{\text{fake}}$ generated by the model, i.e. $\mathbf{x}^{\text{fake}} = \hat{\mathbf{x}}_t = \mathbf{x}_t^{\text{real}} - \boldsymbol{F}_{\boldsymbol{\theta}}(\mathbf{x}_t^{\text{real}}, r) \cdot t$ ($t = 1, r = 0$ are used in implementation, i.e. $\mathbf{x}^{\text{fake}} = \mathbf{z} - \boldsymbol{F}_{\boldsymbol{\theta}}(\mathbf{z}, 0)$), and construct a corresponding "fake trajectory", in which its perturbed version is defined as $\mathbf{x}_{t'}^{\text{fake}} = \alpha(t')\mathbf{z}^{\text{fake}} + \gamma(t')\mathbf{x}^{\text{fake}}$, $\mathbf{z}^{\text{fake}} \sim \mathcal{N}(\mathbf{0}, \mathbf{I})$, and $t' \sim \mathrm{U}(0, 1)$. Here $\mathbf{z}^{\text{fake}}$ is a different noise, which does not need to be the same as $\mathbf{z}$. The network is then trained with the following flow matching objective on this trajectory, using negative time inputs $-t' \in [-1, 0]$:

$$\mathcal{L}(\boldsymbol{\theta})_{\text{adv}} = \mathbb{E}_{\mathbf{x}^{\text{fake}}, \mathbf{z}^{\text{fake}}, t'} \left[ d \left( \boldsymbol{F}_{\boldsymbol{\theta}}(\mathbf{x}_{t'}^{\text{fake}}, -t'), \mathbf{z}^{\text{fake}} - \mathbf{x}^{\text{fake}} \right) \right], \qquad (2)$$

where $d(\cdot, \cdot)$ is a metric function. Minimizing this loss teaches the network to learn the negative time condition and the transformation from noise to fake data distribution, setting the stage for the rectification loss described in the next section.

## 3.2 RECTIFYING REAL TRAJECTORY VIA VELOCITY MATCHING

Ideally, we want the twin trajectories to match with each other. As established in Sec. 3.1, the distributions $p_{\text{fake}}$ and $p_{\text{real}}$ correspond to trajectories parameterized by the negative and positive time intervals, respectively. Inspired by DMD (Yin et al., 2024b), we can treat this as a distribution matching problem. For any perturbed sample $\mathbf{x}_t$, we aim to minimize the KL divergence between these two distributions:

$$D_{\text{KL}}(p_{\text{fake}} \| p_{\text{real}}) = \mathbb{E}_{\mathbf{x}_t, \mathbf{z}, t}\left[\log\left(\frac{p_{\text{fake}}(\mathbf{x}_t)}{p_{\text{real}}(\mathbf{x}_t)}\right)\right] = \mathbb{E}_{\mathbf{x}_t, \mathbf{z}, t}\left[-\left(\log p_{\text{real}}(\mathbf{x}_t) - \log p_{\text{fake}}(\mathbf{x}_t)\right)\right]. \quad (3)$$

**Velocity matching as distribution matching.** Taking the gradient of (3), we derive:

$$\nabla_{\boldsymbol{\theta}} D_{\text{KL}}(p_{\text{fake}} \| p_{\text{real}}) = \nabla_{\boldsymbol{\theta}} \mathbb{E}_{\mathbf{x}_t, \mathbf{z}, t}\left[\log p_{\text{fake}}(\mathbf{x}_t) - \log p_{\text{real}}(\mathbf{x}_t)\right]$$

$$= \mathbb{E}_{\mathbf{x}_t, \mathbf{z}, t}\left[\frac{\partial \log p_{\text{fake}}(\mathbf{x}_t)}{\partial \mathbf{x}_t}\frac{\partial \mathbf{x}_t}{\partial \boldsymbol{\theta}} - \frac{\partial \log p_{\text{real}}(\mathbf{x}_t)}{\partial \mathbf{x}_t}\frac{\partial \mathbf{x}_t}{\partial \boldsymbol{\theta}}\right]$$

$$= \mathbb{E}_{\mathbf{x}_t, \mathbf{z}, t}\left[\Big(\underbrace{\nabla_{\mathbf{x}_t} \log p_{\text{fake}}(\mathbf{x}_t) - \nabla_{\mathbf{x}_t} \log p_{\text{real}}(\mathbf{x}_t)}_{\mathbf{s}_{\text{fake}}(\mathbf{x}_t) - \mathbf{s}_{\text{real}}(\mathbf{x}_t)}\Big)\frac{\partial \mathbf{x}_t}{\partial \boldsymbol{\theta}}\right], \quad (4)$$

where $\mathbf{s}(\cdot)$ is the score of the respective distribution. The relationship between the score and the velocity field $\boldsymbol{F}_{\boldsymbol{\theta}}$ under linear transport ($\alpha(t) = t, \gamma(t) = 1 - t$) is given by (see proof in App. D.1):

$$\mathbf{s}(\mathbf{x}_t) = -\frac{\mathbf{x}_t + \gamma(t) \cdot \boldsymbol{F}_{\boldsymbol{\theta}}(\mathbf{x}_t, t)}{\alpha(t)} = -\frac{\mathbf{x}_t + (1 - t) \cdot \boldsymbol{F}_{\boldsymbol{\theta}}(\mathbf{x}_t, t)}{t}. \quad (5)$$

Substituting this relationship from (5) into the KL gradient (4) yields:

$$\nabla_{\boldsymbol{\theta}} D_{\text{KL}} = \mathbb{E}_{\mathbf{x}_t, \mathbf{z}, t}\left[\left(-\frac{\mathbf{x}_t + (1 - t) \cdot \boldsymbol{F}_{\boldsymbol{\theta}}(\mathbf{x}_t, -t)}{t} - \left(-\frac{\mathbf{x}_t + (1 - t) \cdot \boldsymbol{F}_{\boldsymbol{\theta}}(\mathbf{x}_t, t)}{t}\right)\right)\frac{\partial \mathbf{x}_t}{\partial \boldsymbol{\theta}}\right]$$

$$= \mathbb{E}_{\mathbf{x}_t, \mathbf{z}, t}\left[-\frac{(1 - t)}{t} \cdot \Big(\underbrace{\boldsymbol{F}_{\boldsymbol{\theta}}(\mathbf{x}_t, -t)}_{\mathbf{v}_{\text{fake}}(\mathbf{x}_t, -t)} - \underbrace{\boldsymbol{F}_{\boldsymbol{\theta}}(\mathbf{x}_t, t)}_{\mathbf{v}_{\text{real}}(\mathbf{x}_t, t)}\Big)\frac{\partial \mathbf{x}_t}{\partial \boldsymbol{\theta}}\right], \quad (6)$$

where the model is conditioned on $-t$ for the fake trajectory and on $t$ for the real one. For simplicity, we denote this velocity difference (see Fig. 2a) as:

$$\Delta_{\mathbf{v}}(\mathbf{x}_t) := \mathbf{v}_{\text{real}}(\mathbf{x}_t, t) - \mathbf{v}_{\text{fake}}(\mathbf{x}_t, -t). \quad (7)$$

This derivation recasts the original distribution matching problem into a more practical velocity matching problem. We now show how to formulate this into a tractable rectification loss below.

**Rectification loss derivation.** To derive the rectification loss, we first instantiate the gradient (6) using the setup in Sec. 3.1. In this setting, the network's prediction $\hat{\mathbf{x}}_t$ serves as the clean example, and consequently, the perturbed variable $\mathbf{x}_t$ in (6) corresponds to the fake sample $\mathbf{x}_{t'}^{\text{fake}}$. The velocity difference $\Delta_{\mathbf{v}}(\mathbf{x}_t)$ defined in (7) is therefore instantiated as $\Delta_{\mathbf{v}}(\mathbf{x}_{t'}^{\text{fake}}) = \mathbf{v}_{\text{real}}(\mathbf{x}_{t'}^{\text{fake}}, t') - \mathbf{v}_{\text{fake}}(\mathbf{x}_{t'}^{\text{fake}}, -t')$.

Under this setup, the Jacobian term in (6) is instantiated as $\frac{\partial \mathbf{x}_{t'}^{\text{fake}}}{\partial \boldsymbol{\theta}}$ and simplified to:

$$\frac{\partial \mathbf{x}_{t'}^{\text{fake}}}{\partial \boldsymbol{\theta}} = \frac{\partial(\alpha(t')\mathbf{z}^{\text{fake}} + \gamma(t')\mathbf{x}^{\text{fake}})}{\partial \boldsymbol{\theta}} = \frac{\partial(\alpha(t')\mathbf{z}^{\text{fake}} + \gamma(t')\hat{\mathbf{x}}_t)}{\partial \boldsymbol{\theta}} \propto -\frac{\partial \boldsymbol{F}_{\boldsymbol{\theta}}(\mathbf{x}_t^{\text{real}}, r)}{\partial \boldsymbol{\theta}} \overset{t=1, r=0}{=\!=} -\frac{\partial \boldsymbol{F}_{\boldsymbol{\theta}}(\mathbf{z}, 0)}{\partial \boldsymbol{\theta}}. \quad (8)$$

The KL gradient in (6) thus takes the form of an expectation over the inner product $-\langle\Delta_{\mathbf{v}}(\mathbf{x}_{t'}^{\text{fake}}), \frac{\partial \boldsymbol{F}_{\boldsymbol{\theta}}}{\partial \boldsymbol{\theta}}\rangle$. To construct a tractable loss that produces this gradient structure, we employ the stop-gradient operator, $\text{sg}(\cdot)$. This motivates the following rectification loss:

$$\mathcal{L}(\boldsymbol{\theta})_{\text{rectify}} = \mathbb{E}_{\mathbf{x}_t, \mathbf{x}^{\text{fake}}, \mathbf{z}^{\text{fake}}, t'}\left[d\left(\boldsymbol{F}_{\boldsymbol{\theta}}(\mathbf{z}, 0), \text{sg}\left[\Delta_{\mathbf{v}}(\mathbf{x}_{t'}^{\text{fake}}) + \boldsymbol{F}_{\boldsymbol{\theta}}(\mathbf{z}, 0)\right]\right)\right], \quad (9)$$

where $d(\cdot, \cdot)$ is a metric function. Minimizing $\mathcal{L}_{\text{rectify}}$ encourages the model to straighten the generative trajectories from noise to the data distribution. This rectification allows the entire integration process to be accurately approximated with large step sizes, enabling few-step or 1-step generation.

### 3.3 THE TWINFLOW OBJECTIVE WITH PRACTICAL DESIGNS

**Integration with the any-step framework.** Our method TWINFLOW trains a single model to excel at both multi-step and few-step generation. This is achieved by combining two complementary objectives with conflicting demands:

- The self-adversarial loss ($\mathcal{L}(\boldsymbol{\theta})_{\mathrm{adv}}$ in (2)) promotes high-fidelity, multi-step generation by extending the training dynamics to the interval $t \in [-1, 0]$.
- The rectification loss ($\mathcal{L}(\boldsymbol{\theta})_{\mathrm{rectify}}$ in (9)) optimizes for few-step efficiency by directly straightening the noise-to-data trajectory, enabling rapid, high-quality synthesis.

This creates a dual objective: the model must be both a precise multi-step sampler and an efficient few-step generator. This leads to application of the any-step framework introduced in Sec. 2, which unifies the demands of (2) and (9). We adopt $N = 2$ formulation of (1) to enhance the training stability.

Our final loss combines the base objective with our proposed terms, which we collectively name it $\mathcal{L}(\boldsymbol{\theta})_{\mathrm{TwinFlow}}$. The overall loss function in our methodology can be expressed as:

$$\mathcal{L}(\boldsymbol{\theta}) = \mathcal{L}(\boldsymbol{\theta})_{\mathrm{base}} + (\mathcal{L}(\boldsymbol{\theta})_{\mathrm{adv}} + \mathcal{L}(\boldsymbol{\theta})_{\mathrm{rectify}}) = \mathcal{L}(\boldsymbol{\theta})_{\mathrm{base}} + \mathcal{L}(\boldsymbol{\theta})_{\mathrm{TwinFlow}} . \tag{10}$$

**Practical implementation of mixed loss.** The $\mathcal{L}(\boldsymbol{\theta})_{\mathrm{base}}$ and $\mathcal{L}(\boldsymbol{\theta})_{\mathrm{TwinFlow}}$ objectives in $\mathcal{L}(\boldsymbol{\theta})$ impose different requirements on the target time $r$ under the any-step formulation. Specifically, $\mathcal{L}(\boldsymbol{\theta})_{\mathrm{base}}$ requires $r$ to be sampled from $[0, 1]$, whereas $\mathcal{L}(\boldsymbol{\theta})_{\mathrm{TwinFlow}}$ necessitates a fixed target time of $r = 0$. To accommodate both within a single training step, we partition each mini-batch into two subsets.

A balancing hyperparameter $\lambda$ controls the relative size of these subsets. One portion of the batch is used to compute $\mathcal{L}(\boldsymbol{\theta})_{\mathrm{TwinFlow}}$ with $r = 0$, while the remainder is used for $\mathcal{L}(\boldsymbol{\theta})_{\mathrm{base}}$ with a randomly sampled $r \in [0, 1]$. The value of $\lambda$ thus balances the influence of the two losses on the gradient updates. Setting $\lambda = 0$ disables the $\mathcal{L}(\boldsymbol{\theta})_{\mathrm{TwinFlow}}$ term, while larger values increase its contribution. An ablation study on the impact of $\lambda$ is available in Fig. 4a.

## 4 EXPERIMENTS

We demonstrate the effectiveness of our method, TWINFLOW, on two fronts. First, we highlight its versatility and scalability, we apply TWINFLOW to unified multi-modal models, e.g. Qwen-Image-20B (Wu et al., 2025a), as shown in Tab. 2. Second, we benchmark it against state-of-the-art (SOTA) dedicated text-to-image models, with results presented in Tab. 4.

### 4.1 EXPERIMENTAL SETUP

This section details the experimental setup and evaluation protocol of our proposed methodology.

- **Image generation on multimodal generative models.** We conduct evaluations on unified multimodal models (i.e. takes both texts and images as conditions and capable of generating texts and images). *(1) Network architectures:* We conducted LoRA (Hu et al., 2022) (Tab. 2) and full-parameter training (Tab. 3) of TWINFLOW on Qwen-Image. We also do full-parameter training experiments on OpenUni-512 (Wu et al., 2025c). *(2) Benchmarks:* Following recent works (Pan et al., 2025; Chen et al., 2025b; Deng et al., 2025; Wu et al., 2025a), we use benchmarks in text-to-image generation tasks. For text-to-image generation, we use GenEval, DPG-Bench (Hu et al., 2024), and WISE (Niu et al., 2025). Other training settings are detailed in App. C.1.
- **Text-to-image generation.** For text-to-image generation, we evaluate on dedicated text-to-image models (i.e. primarily takes texts as condition and only generating images). *(1) Network architectures:* We use SANA-0.6B/1.6B (Xie et al., 2024a) in our experiments. *(2) Benchmarks:* Following SANA-series (Xie et al., 2024a; 2025a), we use GenEval (Ghosh et al., 2023) and DPG-Bench (Hu et al., 2024) as evaluation metrics. Other training settings are detailed in App. C.2.

### 4.2 IMAGE GENERATION ON MULTIMODAL GENERATIVE MODELS

We demonstrate TWINFLOW's scalability by achieving competitive 1-NFE text-to-image generation on the *20B-parameter Qwen-Image series* (Wu et al., 2025a). This breakthrough addresses a critical gap in the field, as prior few-step approaches are rarely applied on models exceeding 3B parameters due to instability in GAN-based loss at scale.

Our approach offers two key advantages over state-of-the-art unified multimodal generative models:

(a) TWINFLOW maintains >0.86 GenEval score at 1-NFE on Qwen-Image-20B: surpassing most multi-step models (40-100 NFEs), e.g. Bagel (Deng et al., 2025), MetaQuery (Pan et al., 2025).

Table 2: **System-level comparison of TWINFLOW with unified multimodal models in efficiency and performance on text-to-image tasks.** The **best** and second best results of 1-NFE and 2-NFE are highlighted. † means using LLM rewritten prompts for GenEval. Qwen-Image-TWINFLOW in this table is under LoRA training. Qwen-Image-Lightning⋆ generates almost identical images for the same prompt when evaluating on GenEval and DPG-Bench.

| Method | NFE ↓ | Image Generation | | |
| --- | --- | --- | --- | --- |
| | | GenEval ↑ | DPG-Bench ↑ | WISE ↑ |
| Chameleon (Team, 2024) | - | 0.39 | - | - |
| SEED-X (Ge et al., 2024) | 50×2 | 0.49 | - | - |
| Show-o (Xie et al., 2024b) | 50×2 | 0.68 | 67.27 | 0.35 |
| Janus-Pro (Chen et al., 2025e) | - | 0.80 | 84.19 | 0.35 |
| MetaQuery-XL (Pan et al., 2025) | 30×2 | 0.78 / 0.80† | 81.10 | 0.55 |
| BLIP3-o-8B (Chen et al., 2025b) | 30×2 + 50×2 | 0.84 | 81.60 | 0.62 |
| UniWorld-V1 (Lin et al., 2025) | 28×2 | 0.80 / 0.84† | - | 0.55 |
| OpenUni-L-512 (Wu et al., 2025c) | 20×2 | 0.85 | 81.54 | 0.52 |
| Bagel (Deng et al., 2025) | 50×2 | 0.82 / 0.88† | - | 0.52 |
| Show-o2-7B (Xie et al., 2025b) | 50×2 | 0.76 | 86.14 | 0.39 |
| OmniGen (Xiao et al., 2024) | 50×2 | 0.70 | 81.16 | - |
| OmniGen2 (Wu et al., 2025b) | 50×2 | 0.80 / 0.86† | 83.57 | - |
| Qwen-Image (Wu et al., 2025a) | 50×2 | 0.87 / 0.91RL | 88.32 | 0.62 |
| Qwen-Image-Lightning⋆ (ModelTC, 2025) | 1 | 0.85 | **87.79** | 0.51 |
| OpenUni-RCGM-512 (Sun & Lin, 2025) | 2 | 0.85 | 80.15 | 0.50 |
| OpenUni-RCGM-512 (Sun & Lin, 2025) | 1 | 0.80 | 76.40 | 0.45 |
| **OpenUni-TWINFLOW-512 (Ours)** | 2 | 0.85 | 79.82 | 0.50 |
| | 1 | 0.83 | 79.07 | 0.48 |
| Qwen-Image-RCGM (Sun & Lin, 2025) | 2 | 0.82 | 84.09 | 0.50 |
| Qwen-Image-RCGM (Sun & Lin, 2025) | 1 | 0.52 | 59.50 | 0.30 |
| **Qwen-Image-TWINFLOW (Ours)** | 2 | **0.87 / 0.91†** | 87.64 | **0.57** |
| | 1 | 0.86 / 0.90† | 86.52 | 0.54 |

(b) TWINFLOW achieves this without auxiliary networks, unlike competing few-step methods that require distillation or specialized training pipelines (Yin et al., 2024b;a).

We evaluate the text-to-image generation capabilities of Qwen-Image-TWINFLOW on several standard benchmarks: GenEval (Ghosh et al., 2023), DPG-Bench (Hu et al., 2024), and WISE (Niu et al., 2025). Our model demonstrates strong performance across all benchmarks with only 1-NFE, achieving results that are both competitive and efficient. Detailed results are provided in App. C.1.

**Evaluation on text-to-image benchmarks.** As shown in Tab. 2, Qwen-Image-TWINFLOW achieves a score of 0.86 on GenEval and 86.52% on DPG-Bench with just 1-NFE, *closely matching the original model's performance at 100-NFE.* Compared to Qwen-Image-Lightning (ModelTC, 2025), a 4-step distilled model, our model surpasses it on GenEval and WISE with only 1-NFE. Furthermore, our model outperforms Qwen-Image-RCGM (Sun & Lin, 2025) on both GenEval and DPG-Bench under 1-NFE and 2-NFE settings, with notable improvements of 0.34† on GenEval, 27.0%† on DPG-Bench, and 0.25† on WISE under the 1-NFE setting.

We also benchmark Qwen-Image-TWINFLOW against other prominent multi-step unified multimodal generative models, such as MetaQuery-XL (Pan et al., 2025), BLIP3-o-8B (Chen et al., 2025b), and Bagel (Deng et al., 2025). Our model consistently surpasses these baselines with 1 or 2-NFE across all evaluation metrics. Beyond Qwen-Image, we also apply TWINFLOW to OpenUni (Wu et al., 2025c), achieving GenEval of 0.80 and DPG-Bench of 76.40 under the 1-NFE setting, which is also close to its original performance. These findings underscore the versatility and effectiveness of TWINFLOW across different architectures and scales.

**Further exploration on 20B full-parameter training on Qwen-Image.** Tab. 3 demonstrates the scalability and performance advantages of TWINFLOW on the large-scale Qwen-Image-20B. Existing distribution matching methods, such as VSD (Wang et al., 2023), DMD (Yin et al., 2024b), and SiD (Zhou et al., 2024), typically require maintaining three separate model copies (generator, real score, and fake score), leading to significant memory overhead. In contrast, TWINFLOW distinguishes itself through a unified design:

Table 3: **Comparison of full-parameter training efficiency and performance between TWINFLOW and baselines on text-to-image tasks using Qwen-Image 20B.** The `raw` setting denotes that the generator, real score, and fake score are instantiated as separate models. Though using FSDP-v2, this configuration still leads to OOM. Therefore, for VSD, SiD, and DMD, the fake score is implemented using LoRA ($r = 64$) to ensure memory feasibility. $\star$ indicates severe diversity degradation (mode collapse), characterized by nearly identical outputs on GenEval and DPG-Bench. For sCM and MeanFlow, the JVP is approximated via finite difference.

| Method | NFE ↓ | Image Generation | | |
| | | GenEval ↑ | DPG-Bench ↑ | WISE ↑ |
| --- | --- | --- | --- | --- |
| Qwen-Image (Wu et al., 2025a) | 50×2 | 0.87 / 0.91$^{RL}$ | 88.32 | 0.62 |
| VSD (Wang et al., 2023) (`raw`) | - | OOM | OOM | OOM |
| DMD (Yin et al., 2024b) (`raw`) | - | OOM | OOM | OOM |
| SiD (Zhou et al., 2024) (`raw`) | - | OOM | OOM | OOM |
| VSD (Wang et al., 2023) | 1 | 0.67 | 84.44 | 0.22 |
| | 2 | 0.73 | 86.16 | 0.34 |
| DMD$^{\star}$ (Yin et al., 2024b) | 1 | 0.81 | 84.31 | 0.47 |
| | 2 | 0.80 | 84.08 | 0.46 |
| SiD$^{\star}$ (Zhou et al., 2024) | 1 | 0.77 | 87.05 | 0.42 |
| | 2 | 0.78 | 86.94 | 0.41 |
| sCM (Lu & Song, 2024) (JVP-free) | 1 | 0.55 | 79.12 | 0.21 |
| | 2 | 0.64 | 83.91 | 0.46 |
| MeanFlow (Geng et al., 2025) (JVP-free) | 1 | 0.49 | 82.39 | 0.33 |
| | 2 | 0.57 | 85.09 | 0.40 |
| RCGM (Sun & Lin, 2025) | 1 | 0.56 | 76.15 | 0.31 |
| | 2 | 0.78 | 85.01 | 0.50 |
| **Ours** | 1 | 0.85 | 85.44 | 0.51 |
| | 2 | 0.86 | 86.35 | 0.55 |
| **Ours (longer training)** | 1 | **0.89** | **87.54** | **0.57** |
| | 2 | **0.90** | **87.80** | **0.59** |

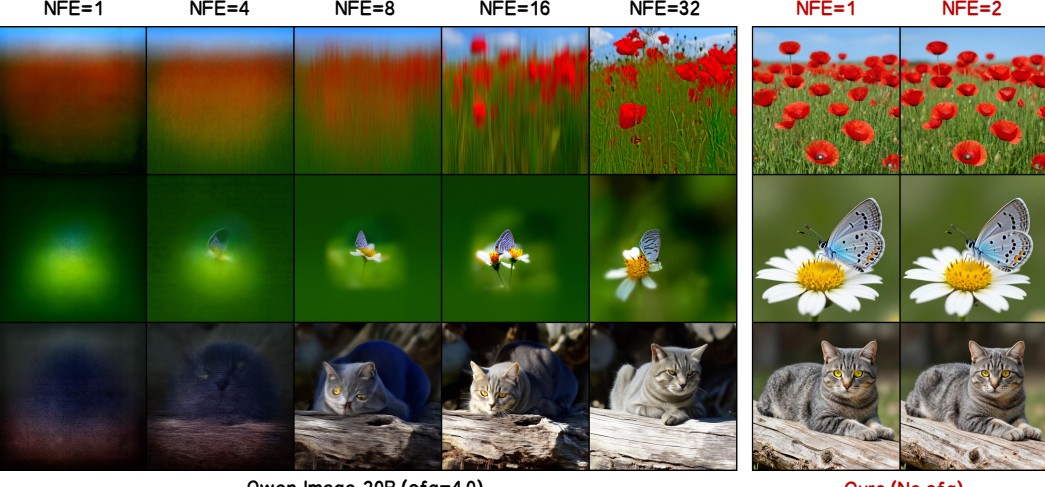

Figure 3: **Visualization of images generated by Qwen-Image and Qwem-Image-TWINFLOW w.r.t. NFEs.** Qwen-Image-TWINFLOW is capable of generating high-quality images with just 1 NFE, which is better than the original Qwen-Image's performance at 16 NFEs. Furthermore, when comparing 2-NFE results to the 32-NFE outputs of Qwen-Image, our method demonstrates better visual details. See prompts in App. E.3.

(a) **Simplicity and efficiency:** By integrating the generator, real/fake score estimation into a single model, TWINFLOW eliminates the need for redundant parameters. This allows for full-parameter training at the 20B scale.

(b) **Performance superiority:** With this unified design, TWINFLOW outperforms all baselines on Qwen-Image-20B. Notably, it achieves superior generation quality with just 1-2 NFE compared to sCM (Lu & Song, 2024) and MeanFlow (Geng et al., 2025).

Table 4: **System-level comparison of TWINFLOW with text-to-image models in efficiency and performance.** Throughput (batch=10) and latency (batch=1) are benchmarked on a single A100 with BF16 precision. The **best** and second best results across 1-NFE are highlighted. † means results tested by ourselves.

| Method | NFE ↓ | Throughput ↑ (samples/s) | Latency (s) ↓ | #Params | GenEval ↑ | DPG-Bench ↑ |
|---|---|---|---|---|---|---|
| **Pretrained multi-step models** | | | | | | |
| SDXL (Podell et al., 2023) | 50×2 | 0.15 | 6.5 | 2.6B | 0.55 | 74.7 |
| PixArt-Σ (Chen et al., 2024a) | 20×2 | 0.40 | 2.7 | 0.6B | 0.54 | 80.5 |
| SD3-Medium (Esser et al., 2024b) | 28×2 | 0.28 | 4.4 | 2.0B | 0.62 | 84.1 |
| FLUX-Dev (Labs, 2024) | 50×2 | 0.04 | 23.0 | 12.0B | 0.67 | 84.0 |
| Playground v3 (Liu et al., 2024) | - | 0.06 | 15.0 | 24B | 0.76 | 87.0 |
| SANA-0.6B (Xie et al., 2024a) | 20×2 | 1.7 | 0.9 | 0.6B | 0.64 | 83.6 |
| SANA-1.6B (Xie et al., 2024a) | 20×2 | 1.0 | 1.2 | 1.6B | 0.66 | 84.8 |
| SANA-1.5 (Xie et al., 2025a) | 20×2 | 0.26 | 4.2 | 4.8B | 0.81 | 84.7 |
| Lumina-Image-2.0 (Qin et al., 2025) | 18×2 | - | - | 2.6B | 0.73 | 87.2 |
| **Few-step models (training *w/* auxiliary models)** | | | | | | |
| SDXL-DMD2 (Yin et al., 2024a) | 2 | 2.89 | 0.40 | 0.9B | 0.58 | - |
| FLUX-Schnell (Labs, 2024) | 2 | 0.92 | 1.15 | 12.0B | 0.71 | - |
| SANA-Sprint-0.6B (Chen et al., 2025c) | 2 | 6.46 | 0.25 | 0.6B | 0.76 | 81.5† |
| SANA-Sprint-1.6B (Chen et al., 2025c) | 2 | 5.68 | 0.24 | 1.6B | 0.77 | 82.1† |
| PixArt-DMD (Chen et al., 2024a) | 1 | 4.26 | 0.25 | 0.6B | 0.45 | - |
| SDXL-DMD2 (Yin et al., 2024a) | 1 | 3.36 | 0.32 | 0.9B | 0.59 | - |
| FLUX-Schnell (Labs, 2024) | 1 | 1.58 | 0.68 | 12.0B | 0.69 | - |
| SANA-Sprint-0.6B (Chen et al., 2025c) | 1 | 7.22 | 0.21 | 0.6B | 0.72 | 78.6† |
| SANA-Sprint-1.6B (Chen et al., 2025c) | 1 | 6.71 | 0.21 | 1.6B | 0.76 | 80.1† |
| **Few-step models (training *w/o* auxiliary models)** | | | | | | |
| SDXL-LCM (Luo et al., 2023) | 2 | 2.89 | 0.40 | 0.9B | 0.44 | - |
| PixArt-LCM (Chen et al., 2024b) | 2 | 3.52 | 0.31 | 0.6B | 0.42 | - |
| PCM (Wang et al., 2024a) | 2 | 2.62 | 0.56 | 0.9B | 0.55 | - |
| SD3.5-Turbo (Esser et al., 2024a) | 2 | 1.61 | 0.68 | 8.0B | 0.53 | - |
| RCGM-0.6B (Sun & Lin, 2025) | 2 | 6.50 | 0.26 | 0.6B | 0.85 | 80.3 |
| RCGM-1.6B (Sun & Lin, 2025) | 2 | 5.71 | 0.25 | 1.6B | 0.84 | 79.1 |
| **TWINFLOW-0.6B (Ours)** | 2 | 6.50 | 0.26 | 0.6B | 0.84 | 79.7 |
| **TWINFLOW-1.6B (Ours)** | 2 | 5.71 | 0.25 | 1.6B | 0.83 | 79.6 |
| SDXL-LCM (Luo et al., 2023) | 1 | 3.36 | 0.32 | 0.9B | 0.28 | - |
| PixArt-LCM (Chen et al., 2024b) | 1 | 4.26 | 0.25 | 0.6B | 0.41 | - |
| PCM (Wang et al., 2024a) | 1 | 3.16 | 0.40 | 0.9B | 0.42 | - |
| SD3.5-Turbo (Esser et al., 2024a) | 1 | 2.48 | 0.45 | 8.0B | 0.51 | - |
| RCGM-0.6B (Sun & Lin, 2025) | 1 | 7.30 | 0.23 | 0.6B | 0.80 | 77.2 |
| RCGM-1.6B (Sun & Lin, 2025) | 1 | 6.75 | 0.22 | 1.6B | 0.78 | 76.5 |
| TiM (Wang et al., 2025) | 1 | - | - | 0.8B | 0.67 | 75.0 |
| **TWINFLOW-0.6B (Ours)** | 1 | 7.30 | 0.23 | 0.6B | **0.83** | 78.9 |
| **TWINFLOW-1.6B (Ours)** | 1 | 6.75 | 0.22 | 1.6B | 0.81 | **79.1** |

**Discussion on open-source community efforts.** To the best of our knowledge, Qwen-Image-Lightning (ModelTC, 2025) is the only open-source few-step model on large models. It is developed using DMD2 (Yin et al., 2024a) but removing GAN loss. This also indirectly reflects the high cost associated with using GAN loss. However, we observe that Qwen-Image-Lightning suffers from severe *mode collapse*: when given the same prompt but different noise inputs, the generated images remain nearly identical across runs. This lack of diversity is empirically demonstrated in the visual comparisons provided in App. E.1.

**Exploration on image editing.** We also conducted a preliminary exploration of our TWINFLOW's capabilities in image editing using a small tuning dataset of approximately 15K editing pairs. Despite the limited scale, our results (see Tab. 8) demonstrate that TWINFLOW can convert Qwen-Image-Edit (Wu et al., 2025a) into a 4-NFE editing model. This suggests that with access to more diverse editing datasets, we anticipate further improvements in both fidelity and versatility of edited outputs.

## 4.3 IMAGE GENERATION ON DEDICATED TEXT-TO-IMAGE MODELS

To validate our method's versatility, we also benchmark it on traditional text-to-image generation. As shown in Tab. 4, we first benchmark against pretrained multi-step models (typically requiring >40-NFE). Following the categorization in Tab. 1, we compare against SOTA few-step models, grouped by their reliance on auxiliary components: those trained with versus without auxiliary models. Critically, full-parameter tuning on SANA-0.6B/1.6B backbones enables high-fidelity image generation in just 1-2 NFE.

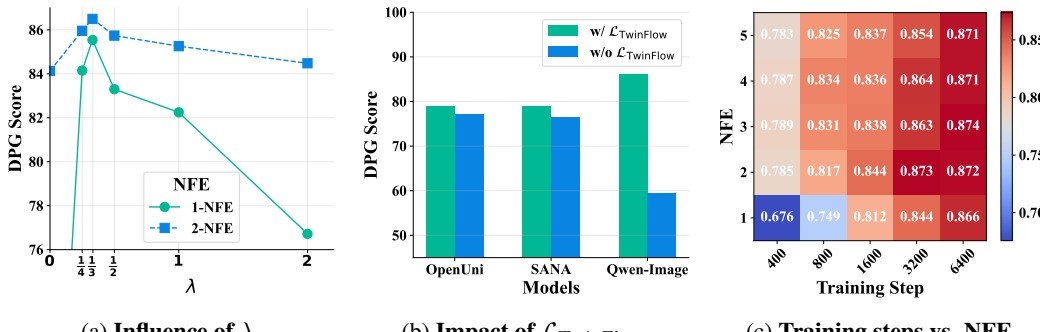

(a) **Influence of $\lambda$.**      (b) **Impact of $\mathcal{L}_{\text{TwinFlow}}$.**      (c) **Training steps vs. NFE.**

Figure 4: **Ablation studies of TWINFLOW.** Ablation presented in (a) and (c) are conducted on Qwen-Image-TWINFLOW. Results shown in (b) are trained on the same dataset but with different models.

(a) **1-NFE setting:** The efficacy of our method is particularly pronounced in the more demanding 1-NFE inference setting. Here, our models (0.6B: 0.83, 1.6B: 0.81 on GenEval) significantly outperform other leading 1-NFE methods, such as SANA-RCGM (0.78) (Sun & Lin, 2025), SANA-Sprint (0.76) (Chen et al., 2025c), FLUX-Schnell (0.69) (Labs, 2024), and SDXL-DMD2 (0.59) (Yin et al., 2024a). Notably, our 1-NFE TWINFLOW-0.6B (GenEval: 0.83) exceeds the generation quality of the 40-NFE SANA-1.5-4.8B (Xie et al., 2025a) model while offering substantially greater computational efficiency.

(b) **2-NFE setting:** TWINFLOW-0.6B achieves a throughput of 6.50 samples/s and a latency of 0.26s, performance metrics comparable to the originally reported SANA values. On the GenEval benchmark, our model attains a score of 0.84, surpassing not only the SANA-Sprint series (0.76 and 0.77) but also powerful multi-step models like SANA-1.5 (0.81) and Playground v3 (0.76). Our models also demonstrate competitive performance on DPG-Bench, with scores of 79.7 for the 0.6B variant and 79.6 for the 1.6B variant.

TWINFLOW-0.6B/1.6B achieves SoTA text-to-image generation performance on the GenEval benchmark using just 1-NFE, surpassing both SANA-Sprint and RCGM. While we slightly underperform on DPG-Bench relative to SANA-Sprint, due to SANA-Sprint's reliance on extensive, proprietary training data. We believe this gap is primarily data-driven and can be effectively closed by training on larger, higher-quality datasets.

## 4.4 ABLATION STUDY AND ANALYSIS

**Influence of $\lambda$.** As described in Sec. 3.3, $\lambda$ is designed to control the sample distribution of $\mathcal{L}_{\text{base}}$ and $\mathcal{L}_{\text{TwinFlow}}$. In Fig. 4a, we visualize the DPG-Bench performance w.r.t. $\lambda$ at 1-NFE and 2-NFE. We observed that as $\lambda$ increases from 0, the performance on DPG-Bench initially increases and then decreases, reaching its peak at approximately $\lambda = 1/3$. These results indicate that appropriately balancing samples in the local batch helps improve the model performance.

**Impact of $\mathcal{L}_{\text{TwinFlow}}$ on different models.** We conduct an ablation study to analyze the impact on text-to-image performance of using $\mathcal{L}_{\text{TwinFlow}}$ on different models. As illustrated in Fig. 4b, incorporating $\mathcal{L}_{\text{TwinFlow}}$ significantly enhances performance: it improves 1-NFE performance for the text-to-image task across OpenUni, SANA, and especially Qwen-Image (from 59.50 to 86.52).

**Effect of training steps vs. NFE.** As illustrated in Fig. 4c, the experimental results demonstrate that as training progresses, the "comfort regime" for optimal sampling steps shifts accordingly. Notably, GenEval performance improvements are observed across both 1-step and few-step scenarios, with significant gains achieved as training progresses, which shows the effectiveness of $\mathcal{L}_{\text{TwinFlow}}$.

## 5 CONCLUSION AND LIMITATIONS

We introduce TWINFLOW, a streamlined framework for training large-scale, few-step continuous generative models. Unlike complex DMD-series approaches, our method eliminates auxiliary components like GAN discriminators or frozen teacher models, enabling straightforward and efficient 1-step training. While extensive experiments confirm its high-quality text-to-image synthesis, two primary limitations remain: its scalability to tasks like image editing is unexplored, and its performance across diverse modalities like video and audio requires further validation. Addressing these gaps will be key to developing more versatile generative models.

ACKNOWLEDGEMENT

This work was supported in part by the National Science and Technology Major Project (No. 2022ZD0115101), NSFC under No. 62576285, the Research Center for Industries of the Future (RCIF) at Westlake University, and the Westlake Education Foundation.

We thank the anonymous reviewers for their insightful comments and suggestions, which greatly enhanced the quality of this paper. We also extend our gratitude to Deyuan Liu, Chuyan Chen, and Jun Xie for their helpful discussions and feedback.

## 6 ETHICS STATEMENT

Our work is conducted in full alignment with the *ICLR Code of Ethics*, and we are committed to upholding the principles of transparent and responsible research. This study did not involve human participants or the use of personal or sensitive data, thereby negating the need for an institutional ethics review. All datasets utilized are publicly available under their respective licenses, and we have provided appropriate attribution to their original sources. To foster transparency and enable further innovation, our implementation code and experimental configurations will be made available. We also affirm that no conflicts of interest or external funding have influenced this work.

## 7 REPRODUCIBILITY STATEMENT

To ensure that our findings can be accurately and transparently replicated, we have provided a comprehensive account of our experimental methodology. Exhaustive details concerning the model architectures and evaluation protocols are documented in Sec. 4 of the main text and further elaborated in App. C. Following the acceptance of this paper, we will make our entire source code publicly available to facilitate verification and future research.

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

CONTENTS

## A  USE OF LLMS

This paper only uses LLMs for polishing.

## B  RELATED WORK

**Multi-step generative methods.**  Diffusion (Ho et al., 2020; Dhariwal & Nichol, 2021) and flow matching (Lipman et al., 2022) models have shown impressive performance in image generation. They progressively transport a simple noise distribution to the data distribution, either by reversing a noising process or integrating a probability-flow ODE. However, this iterative nature introduces bottleneck in inference efficiency, as generating an image requires numerous sequential evaluation steps.

**Few-step generative methods.**  Various methods tend to accelerate the sampling process, such as diffusion distillation (Luhman & Luhman, 2021; Salimans & Ho, 2022) and consistency distillation (Song et al., 2023; Song & Dhariwal, 2023). They typically train a student model to approximate the ODE sampling trajectory of a frozen teacher model in fewer sampling steps. While effective at moderate NFEs, these approaches depend on a frozen teacher, and quality often drops sharply in the extreme few-step regime (< 4 NFEs). Incorporating GAN-like loss into distillation (e.g., CTM (Kim et al., 2023), ADD/LADD (Sauer et al., 2024b;a), DMD/DMD2 (Yin et al., 2024b;a)) can improve sharpness and alignment at few steps. Yet these frameworks increase training complexity and instability: they typically introduce auxiliary modules (discriminators, fake-sample score functions) and still rely on a frozen teacher, leading to higher memory overhead and sensitivity to hyperparameters. For ultra-large models, this added complexity often translates to out-of-memory failures or brittle training dynamics.

**Few-step applications in large generative models.**  The tension between speed and quality is amplified in large-scale systems. For instance, Qwen-Image-20B (Wu et al., 2025a) typically requires 100 NFEs, leading to substantial latency (∼40s on a single A100 for 1024×1024 resolution). Recent works distill to cut NFEs while preserving compositionality and aesthetics: LCM-style distillation for latent models (Luo et al., 2023), large-scale text-to-image distillation pipelines (e.g., PixArt-Delta (Chen et al., 2024b), SDXL distillation (Lin et al., 2024), FLUX-schnell (Labs, 2024)), and hybrid frameworks such as SANA-Sprint (Chen et al., 2025c) that combine teacher guidance with adversarial signals. Recently, Hunyuan-Image-2.1 (Team, 2025) explore MeanFlow (Geng et al., 2025) for mid-step acceleration (16 NFEs). Nevertheless, when targeting 1-2 NFEs on 1024×1024 text-to-image with 10B-20B backbones, these pipelines face practical barriers: dependence on frozen teachers, extra discriminators or score networks, unstable adversarial training, and prohibitive memory costs that hinder straightforward scaling.

**Concurrent progress in few-step generative models.**  Recent research has significantly diversified the landscape of efficient generative models, introducing novel training objectives and distillation paradigms. FACM (Peng et al., 2025) stabilizes consistency training by explicitly anchoring the shortcut objective to the instantaneous velocity field of a flow matching task. SoFlow (Luo et al., 2025) proposes a solution consistency loss that bypasses expensive JVP calculations, enabling efficient one-step generation. Adopting a policy-optimization perspective, $\pi$-Flow (Chen et al., 2025a) decouples the number of network evaluations from ODE integration steps via an imitation learning framework termed $\pi$-ID. Furthermore, Tong et al. (2025) introduces a data-free distillation paradigm that samples from the prior distribution to eliminate teacher-data mismatch, while Self-E (Yu et al., 2025) employs a self-evaluation mechanism to refine generation quality by leveraging the model's own estimates. T2I-Distill (Pu et al., 2025) systematically evaluates sCM (Lu & Song, 2024) and MeanFlow (Geng et al., 2025) methods and developed efficient JVP kernels, offering practical guidelines for adapting them to open-domain text-to-image generation.

In the realm of methods with distribution matching, Decoupled-DMD (Liu et al., 2025a) and DMD-R (Jiang et al., 2025) offer new insights into the DMD objective; the former identifies CFG augmentation as the primary driver of acceleration while treating distribution matching as a regularizer, whereas the latter integrates reinforcement learning to enhance mode coverage during distillation. Targeting large-scale video generation, rCM (Zheng et al., 2025) combines continuous-time consistency models with score distillation regularization to resolve fine-detail degradation, and Transition Matching Distillation (TMD) (Nie et al., 2026) utilizes a decoupled architecture with a recurrent flow head to efficiently distill video diffusion models into few-step generators.

## C  DETAILED EXPERIMENTS

### C.1  DETAILED IMPLEMENTATION ON MULTIMODAL GENERATIVE MODELS

**Training Datasets.**  For training OpenUni (Wu et al., 2025c) and Qwen-Image (Wu et al., 2025a), we used the same datasets as in our text-to-image experiments but excluded ShareGPT-4o-Image (Chen et al., 2025d). For exploration on full-parameter training on Qwen-Image, since DMD (Yin et al., 2024b) employs a regression loss, we need to pre-generate offline data using Qwen-Image. To ensure a fair comparison with established baselines, we randomly sample 50K prompts from the datasets we used, and generate images using Qwen-Image. For Qwen-Image-Edit (Wu et al., 2025a), we used only the part of editing split of the ShareGPT-4o-Image dataset.

**Training configurations.**  We performed full-parameter fine-tuning experiments on OpenUni, using an image resolution of $512 \times 512$. With a batch size of 128, the model was trained for 600,000 steps. For other training configurations, please refer to Tab. 6.

For LoRA fine-tuning on Qwen-Image and Qwen-Image-Edit. We set the rank ($r$) and alpha ($\alpha$) to 64 for Qwen-Image, and to 64 and 32, respectively, for Qwen-Image-Edit. This LoRA setup comprises approximately 420M trainable parameters. A comprehensive list of training configuration is provided in Tab. 6. For full-parameter training on Qwen-Image, we add a additional time embedder to serve as the target timestep condition.

**Additional detailed results on GenEval.**  As detailed in Tab. 5, our Qwen-Image-TWINFLOW achieves 0.86 with 1-NFE. Notably, when using LLM rewritten prompts, our GenEval score comes to 0.90, which is very close to Qwen-Image-RL (Wu et al., 2025a) (0.91). Significant score increases are observed in the Colors and Attribute Binding subtasks. This indicates that our model exhibits enhanced image generation capabilities when processing long input instructions.

Table 5: **Detailed evaluation results on GenEval for text-to-image models.** Qwen-Image-Lightning are evaluated with 1-NFE. $^\dagger$means using LLM rewritten prompts for GenEval.

| Model | Single Object | Two Object | Counting | Colors | Position | Attribute Binding | Overall↑ |
|---|---|---|---|---|---|---|---|
| SEED-X (Ge et al., 2024) | 0.97 | 0.58 | 0.26 | 0.80 | 0.19 | 0.14 | 0.49 |
| Emu3-Gen (Wang et al., 2024b) | 0.98 | 0.71 | 0.34 | 0.81 | 0.17 | 0.21 | 0.54 |
| JanusFlow (Ma et al., 2025) | 0.97 | 0.59 | 0.45 | 0.83 | 0.53 | 0.42 | 0.63 |
| Show-o (Xie et al., 2024b) | 0.99 | 0.80 | 0.66 | 0.84 | 0.31 | 0.50 | 0.68 |
| OmniGen (Xiao et al., 2024) | 0.98 | 0.84 | 0.66 | 0.74 | 0.40 | 0.43 | 0.68 |
| Janus-Pro-7B (Chen et al., 2025e) | 0.99 | 0.89 | 0.59 | 0.90 | 0.79 | 0.66 | 0.80 |
| OpenUni-512 (Wu et al., 2025c) | 0.99 | 0.91 | 0.77 | 0.90 | 0.75 | 0.76 | 0.85 |
| Bagel (Deng et al., 2025) | 0.99 | 0.94 | 0.81 | 0.88 | 0.64 | 0.63 | 0.82 |
| OmniGen2 (Wu et al., 2025b) | 1.00 | 0.95 | 0.64 | 0.88 | 0.55 | 0.76 | 0.80 |
| Show-o2-7B (Xie et al., 2025b) | 1.00 | 0.87 | 0.58 | 0.92 | 0.52 | 0.62 | 0.76 |
| Qwen-Image (Wu et al., 2025a) | 0.99 | 0.92 | 0.89 | 0.88 | 0.76 | 0.77 | 0.87 |
| Qwen-Image-Lightning (ModelTC, 2025) | 0.99 | 0.89 | 0.85 | 0.87 | 0.75 | 0.76 | 0.85 |
| **OpenUni-TWINFLOW-512 (1-NFE)** | 0.99 | 0.91 | 0.69 | 0.90 | 0.79 | 0.72 | 0.83 |
| **Qwen-Image-TWINFLOW (1-NFE)** | 1.00 | 0.91 | 0.84 | 0.90 | 0.75 | 0.74 | 0.86 |
| **Qwen-Image-TWINFLOW $^\dagger$ (1-NFE)** | 0.99 | 0.94 | 0.87 | 0.96 | 0.78 | 0.83 | 0.90 |

### C.2  DETAILED IMPLEMENTATION ON DEDICATED TEXT-TO-IMAGE MODELS

**Training datasets.**  Considering training datasets, we use BLIP-3o-60K (Chen et al., 2025b), Echo-4o (w/o multi-reference split) (Ye et al., 2025), and ShareGPT-4o-Image (Chen et al., 2025d). Together, these three instruction tuning datasets comprise approximately 200,000 text-to-image samples.

**Training configurations.**  All experiments on SANA-0.6B/1.6B backbones are conducted with full-parameter tuning. we fine-tuned these two models for 30,000 steps, using batch sizes of 128 and 64 respectively. Other detailed training configurations are provided in Tab. 6.

**Additional detailed results on GenEval.**  We list the detailed results of TWINFLOW-0.6B/1.6B on GenEval (Ghosh et al., 2023) along with other SOTA multi-step text-to-image models (except FLUX-Schnell) in Tab. 7. It demonstrates except the overall score, our TWINFLOW-0.6B/1.6B also

Table 6: **Detailed training configurations for experiments on text-to-image models and multimodal generative models.**

| Configuration | SANA-0.6B | SANA-1.6B | OpenUni-512 | Qwen-Image (LoRA) | Qwen-Image (Full) | Qwen-Image-Edit |
|---|---|---|---|---|---|---|
| **Optimizer Settings** | | | | | | |
| Optimizer | RAdam | RAdam | AdamW | AdamW | AdamW | AdamW |
| Learning Rate | $1 \times 10^{-4}$ | $1 \times 10^{-4}$ | $1 \times 10^{-4}$ | $1 \times 10^{-4}$ | $1 \times 10^{-5}$ | $1 \times 10^{-4}$ |
| Weight Decay | 0 | 0 | 0 | 0 | 0 | 0 |
| $(\beta_1, \beta_2)$ | (0.9, 0.95) | (0.9, 0.99) | (0.9, 0.95) | (0.9, 0.99) | (0.9, 0.99) | (0.9, 0.99) |
| **Training Details** | | | | | | |
| Batch Size | 128 | 128 | 128 | 64 | 32 | 24 |
| Training Steps | 30000 | 30000 | 60000 | 7000 | 3000, 10000 (longer training) | 7000 |
| Learning Rate Scheduler | Constant | Constant | Constant | Constant | Constant | Constant |
| Gradient Clipping | - | - | - | 1.0 | 1.0 | 1.0 |
| Random Seed | 42 | 42 | 42 | 42 | 42 | 42 |
| LoRA $r$ | - | - | - | 64 | - | 64 |
| LoRA $\alpha$ | - | - | - | 64 | - | 32 |
| EMA Decay Rate | 0.99 | 0.99 | 0.99 | 0 | 0.99 | 0 |

outperforms these multi-step models with 1-NFE in sub tasks such as Position (0.6B: 0.84, 1.6B: 0.79) and Attribute Binding (0.6B: 0.70, 1.6B: 0.68).

Table 7: **Detailed evaluation results on GenEval for text-to-image models.**

| Model | Single Object | Two Object | Counting | Colors | Position | Attribute Binding | Overall↑ |
|---|---|---|---|---|---|---|---|
| SDXL (Lin et al., 2024) | 0.98 | 0.74 | 0.39 | 0.85 | 0.15 | 0.23 | 0.55 |
| PixArt-$\Sigma$ (Chen et al., 2024a) | 0.98 | 0.59 | 0.50 | 0.80 | 0.10 | 0.15 | 0.52 |
| SD3-Medium (Esser et al., 2024a) | 0.98 | 0.74 | 0.63 | 0.67 | 0.34 | 0.36 | 0.62 |
| FLUX-Dev (Labs, 2024) | 0.98 | 0.81 | 0.74 | 0.79 | 0.22 | 0.45 | 0.66 |
| FLUX-Schnell (Labs, 2024) | 0.99 | 0.92 | 0.73 | 0.78 | 0.28 | 0.54 | 0.71 |
| SD3.5-Large (Esser et al., 2024a) | 0.98 | 0.89 | 0.73 | 0.83 | 0.34 | 0.47 | 0.71 |
| Lumina-Image-2.0 (Qin et al., 2025) | - | 0.87 | 0.67 | - | - | 0.62 | 0.73 |
| SANA-0.6B (Xie et al., 2024a) | 0.99 | 0.76 | 0.64 | 0.88 | 0.18 | 0.39 | 0.64 |
| SANA-1.6B (Xie et al., 2024a) | 0.99 | 0.77 | 0.62 | 0.88 | 0.21 | 0.47 | 0.66 |
| **TWINFLOW-0.6B (1-NFE)** | 0.98 | 0.90 | 0.68 | 0.89 | 0.84 | 0.70 | 0.83 |
| **TWINFLOW-1.6B (1-NFE)** | 0.99 | 0.88 | 0.65 | 0.86 | 0.79 | 0.68 | 0.81 |

## C.3 EXPLORATION ON IMAGE EDITING

We conducted a preliminary exploration of our method on image editing tasks using a subset of approximately 15,000 square images from Chen et al. (2025d). We fine-tuned the Qwen-Image-Edit model (Wu et al., 2025a) using LoRA at a fixed 512×512 resolution, with training configurations detailed in Table 6. Note that we use a low and fixed resolution during training; during testing, we use a resolution that is the same as the input image size. Our Qwen-Image-Edit-TWINFLOW can effectively edit images in 2 to 4 NFEs. With 2 NFEs, it achieves a score of 3.47 on the ImgEdit, surpassing all multi-step models except Qwen-Image-Edit itself.

Despite the naive usage of the dataset and training strategy, these results highlight the significant potential of our approach. We believe that with further scaling, our method can achieve strong performance in a single-step (1-NFE) setting on image editing tasks.

## D THEORETICAL ANALYSIS

### D.1 TRANSFORMATION FROM SCORE TO VELOCITY

In this section, we derive the equation between score and velocity. According to Equation 2 in Sun et al. (2025) with flow matching, we have:

$$\boldsymbol{f}^{\mathbf{x}}(\boldsymbol{F}_{\boldsymbol{\theta}}(\mathbf{x}_t, t), \mathbf{x}_t, t) = \frac{\alpha(t) \cdot \boldsymbol{F}_{\boldsymbol{\theta}}(\mathbf{x}_t, t) - \hat{\alpha}(t) \cdot \mathbf{x}_t}{\alpha(t) \cdot \hat{\gamma}(t) - \hat{\alpha}(t) \cdot \gamma(t)} = \frac{t \cdot \boldsymbol{F}_{\boldsymbol{\theta}}(\mathbf{x}_t, t) - \mathbf{x}_t}{(t \cdot (-1) - 1 \cdot (1 - t))}$$

$$= \frac{t \cdot \boldsymbol{F}_{\boldsymbol{\theta}}(\mathbf{x}_t, t) - \mathbf{x}_t}{(-1)} = \mathbf{x}_t - t \cdot \boldsymbol{F}_{\boldsymbol{\theta}}(\mathbf{x}_t, t). \tag{11}$$

Table 8: **Comparison of TWINFLOW with unified multimodal models in performance on image editing tasks.** In GEdit-Bench, G_SC measures Semantic Consistency, G_PQ evaluates Perceptual Quality, and G_O reflects the Overall Score. All metrics are evaluated by GPT-4.1.

| Method | NFE ↓ | Image Editing GEdit-EN (Full set) ↑ | | | ImgEdit ↑ |
| | | G_SC | G_PQ | G_O | |
|---|---|---|---|---|---|
| UniWorld-V1 (Lin et al., 2025) | 28×2 | 4.93 | 7.43 | 4.85 | 3.26 |
| OmniGen (Xiao et al., 2024) | 50×2 | 5.96 | 5.89 | 5.06 | 2.96 |
| OmniGen2 (Wu et al., 2025b) | 50×2 | 7.16 | 6.77 | 6.41 | 3.44 |
| Step1X-Edit (Liu et al., 2025b) | 28×2 | 7.66 | 7.35 | 6.97 | 3.06 |
| Bagel (Deng et al., 2025) | 50×2 | 7.36 | 6.83 | 6.52 | 3.20 |
| Qwen-Image (Wu et al., 2025a) | 50×2 | 8.00 | 7.86 | 7.56 | 4.27 |
| **Qwen-Image-Edit-TWINFLOW** | 4 | 5.95 | 6.97 | 5.91 | 3.55 |
| **Qwen-Image-Edit-TWINFLOW** | 2 | 5.94 | 6.81 | 5.85 | 3.47 |

According to Theorem 2 in Sun et al. (2025), we have:

$$f^{\mathbf{x}}(\boldsymbol{F}_{\boldsymbol{\theta}}(\mathbf{x}_t, t), \mathbf{x}_t, t) = \frac{\mathbf{x}_t + \alpha^2(t)\, \nabla_{\mathbf{x}_t} \log p_t(\mathbf{x}_t)}{\gamma(t)} = \frac{\mathbf{x}_t + t^2\, \nabla_{\mathbf{x}_t} \log p_t(\mathbf{x}_t)}{(1-t)} = \frac{\mathbf{x}_t + t^2\, \mathbf{s}(\mathbf{x}_t)}{1-t},$$
(12)

where $\mathbf{s}(\mathbf{x}_t) = \nabla_{\mathbf{x}_t} \log p_t(\mathbf{x}_t)$. By simple transposition of $\mathbf{s}(\mathbf{x}_t)$ term, we can get:

$$\frac{\mathbf{x}_t + t^2\, \mathbf{s}(\mathbf{x}_t)}{1-t} = \mathbf{x}_t - t \cdot \boldsymbol{F}_{\boldsymbol{\theta}}(\mathbf{x}_t, t)$$

$$\implies t^2 \mathbf{s}(\mathbf{x}_t) = (1-t)\,(\mathbf{x}_t - t \cdot \boldsymbol{F}_{\boldsymbol{\theta}}(\mathbf{x}_t, t)) - \mathbf{x}_t = -t\mathbf{x}_t + (t^2 - t)\boldsymbol{F}_{\boldsymbol{\theta}}(\mathbf{x}_t, t)$$

$$\implies \mathbf{s}(\mathbf{x}_t) = \frac{-t\mathbf{x}_t + (t^2 - t)\boldsymbol{F}_{\boldsymbol{\theta}}(\mathbf{x}_t, t)}{t^2} = \boxed{-\frac{\mathbf{x}_t + (1-t)\boldsymbol{F}_{\boldsymbol{\theta}}(\mathbf{x}_t, t)}{t}}.$$
(13)

# E   VISUALIZATION RESULTS

## E.1   COMPARISON OF QWEN-IMAGE-TWINFLOW AND QWEN-IMAGE-LIGHTNING

As Fig. 5 shown, our comparative analysis reveals a notable limitation in Qwen-Image-Lightning's generation. The model produces images with very low diversity; outputs are often highly similar and visually repetitive even when initialized with different latent noise. Our Qwen-Image-TWINFLOW does not exhibit model collapse, demonstrating the ability to generate a rich variety of high-quality images. We also quantify the similarity by using LPIPS metric in Tab. 9. We perform the diversity evaluation on GenEval. Specifically, for each prompt, we compute the average pairwise LPIPS distance among the 4 generated samples. The final score is the mean of all scores across the entire samples. The results quantify that Qwen-Image-Lightning shows obvious diversity degradation.

Table 9: **Quantification of diversity using LPIPS score for Qwen-Image-Lightning and Qwen-Image-TWINFLOW.** Qwen-Image-Lightning shows obvious diversity degradation.

| Model | NFE | LPIPS ↑ |
|---|---|---|
| Qwen-Image-Lightning | 1 | 0.2996 |
| **Qwen-Image-TWINFLOW** | 1 | **0.5044** |
| Qwen-Image-Lightning | 2 | 0.3046 |
| **Qwen-Image-TWINFLOW** | 2 | **0.5188** |

## E.2   VISUALIZATION RESULTS ACROSS TRAINING STEPS

As illustrated in Fig. 6, our method exhibits a two-stage training dynamic. Initially (200 to 400 steps), it demonstrates rapid convergence in 1-NFE performance, quickly establishing a strong baseline.

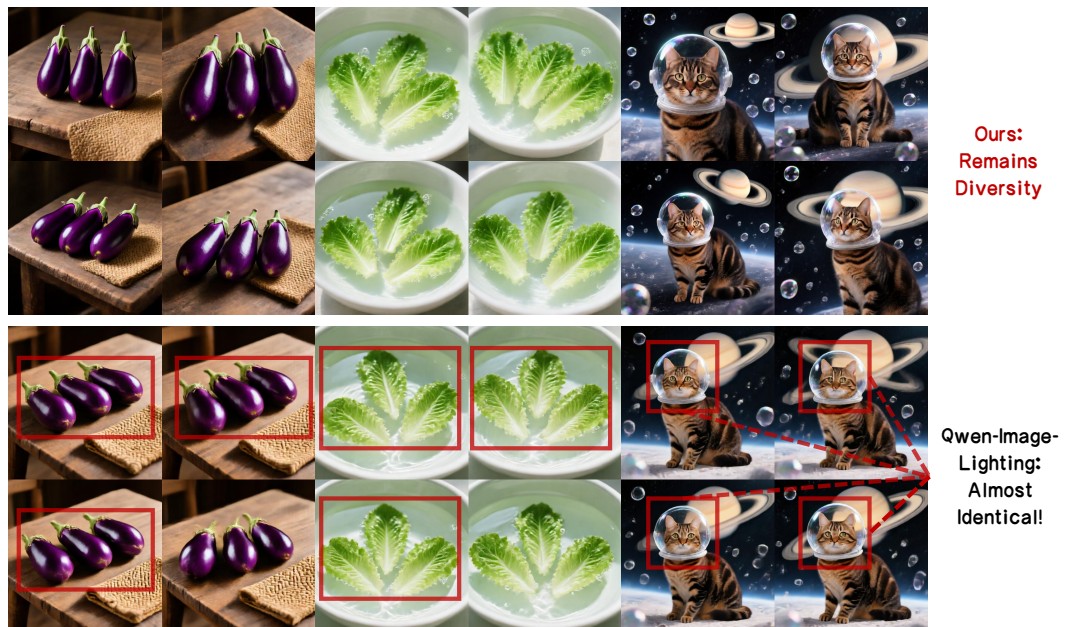

Figure 5: **Comparison between Qwen-Image-TWINFLOW and Qwen-Image-Lightning (1-NFE).** The prompts and generated images are sourced from DPG-Bench. We observe that Qwen-Image-Lightning tend to generate very similar images though noise is different, which hurts diversity. Our model remains diversity and high quality generation.

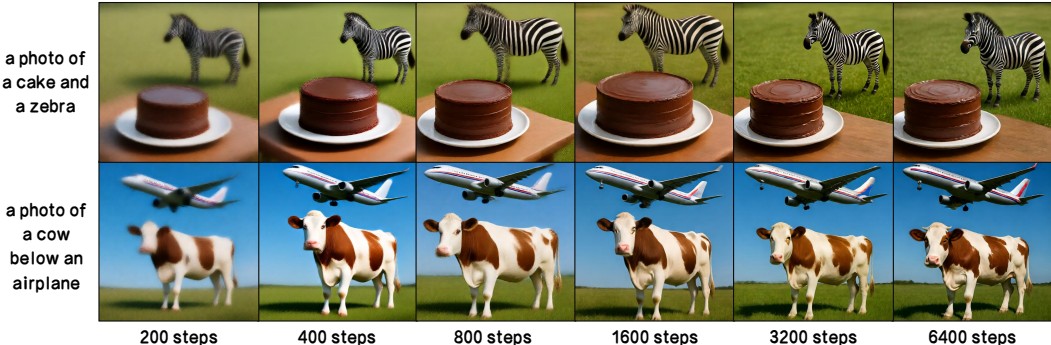

Figure 6: **Visualizations of 1-NFE images generated by Qwen-Image-TWINFLOW w.r.t. training steps.** In the early stages of training, our method converges rapidly, and the generated images begin to take shape (200 to 400 steps); as training progresses, our method gradually optimize the visual details (800 to 6400 steps).

Subsequently (800 to 6400 steps), the training process shifts towards refining finer visual details, leading to a steady enhancement of image fidelity. This highlights our approach's efficiency in achieving strong initial results and its capacity for continued improvement with extended training.

### E.3 SELECTED PROMPTS USED FOR VISUALIZATION

The prompts used to generate the results shown in Fig. 1 and Fig. 3 are detailed in this section to ensure reproducibility.

1. A cinematic vertical composition of a bustling street in Central Hong Kong, featuring vibrant red taxis driving along the road. The scene is framed by towering modern skyscrapers with reflective glass facades, creating an urban jungle atmosphere. The cityscape glows with neon signs and soft ambient light, capturing the essence of Hong Kong's iconic night–time energy. On the road surface, painted traffic markings texts: 'SLOW'. The sky above has a gradient transitioning from deep twilight blue to warm orange hues near the horizon, adding depth and drama to the image. Rendered in hyper–realistic style with rich colors, intricate textures, and high contrast lighting for maximum impact.

2. Classic Baroque–style still life painting, a woven wicker basket overflowing with fresh fruits including red and green grapes, ripe apples, plums, quinces, and a yellow pear, adorned with grape leaves and vines. The basket sits on a draped stone or wooden table covered with a dark blue cloth, with scattered fruits and berries around it. Rich, dramatic lighting highlights the textures and colors of the fruit, creating deep shadows and soft highlights. A luxurious red curtain drapes in the background, adding depth and contrast. Realistic, highly detailed, oil painting style, reminiscent of 17th–century Dutch or Flemish masters such as Jan Davidsz de Heem or Caravaggio. Warm, earthy tones, meticulous attention to detail, and a sense of abundance and natural beauty. Ultra HD, 4K, cinematic composition.

3. Clean white brick wall, vibrant colorful spray–paint graffiti covering entire surface: top giant bubble letters '1 Step Generation', below stacked 'TwinFlow', 'Made Easy' in rainbow palette, fresh wet paint drips, daylight urban photography, realistic light and shadow. Ultra HD, 4K, cinematic composition.

4. A close–up realistic selfie of three cats of different breeds in front of the iconic Big Ben, each with a different expression, taken during the blue hour with cinematic lighting. The animals are close to the camera, heads touching, mimicking a selfie pose, displaying joyful, surprised, and calm expressions. The background showcases the complete architectural details of the [landmark], with soft light and a warm atmosphere. Shot in a photorealistic, cartoon style with high detail.

5. Starbucks miniature diorama shop. The roof is made of oversized coffee beans, and above the windows is a huge 'Starbucks' sign. A vendor is handing coffee to customers, and the ground is covered with many coffee beans. Handmade polymer clay sculpture, studio macro photograph, soft lighting, shallow depth of field. Ultra HD, 4K, cinematic composition.

6. A whimsical scene featuring a capybara joyfully riding a sleek, modern rocket. The capybara is holding a sign with both hands, the text on the sign boldly and eye–catchingly reading 'QWEN–IMAGE–20B'. The capybara looks thrilled, sporting a playful grin as it soars through a vibrant sky filled with soft, pastel clouds and twinkling stars. The rocket leaves a trail of sparkling, colorful smoke behind it, adding to the magical atmosphere. Ultra HD, 4K, cinematic composition.A still frame from a black and white movie, featuring a man in classic attire, dramatic high contrast lighting, deep shadows, retro film grain, and a nostalgic cinematic mood. Ultra HD, 4K, cinematic composition.

7. Close–up portrait of a young woman with light skin and long brown hair, looking directly at the camera. Her face is illuminated by dramatic, slatted sunlight casting shadows across her features, creating a pattern of light and shadow. Her eyes are a striking green, and her lips are slightly parted, with a natural pink hue. The background is a soft, dark gradient, enhancing the focus on her face. The lighting is warm and golden. Ultra HD, 4K, cinematic composition.

8. A field of vibrant red poppies with green stems under a blue sky.

9. A small blue–gray butterfly with black stripes rests on a white and yellow flower against a blurred green background.

10. A grey tabby cat with yellow eyes rests on a weathered wooden log under bright sunlight

### E.4 HIGH RESOLUTION VISUALIZATION

In this section, we showcase further qualitative results from Qwen-Image-TWINFLOW to highlight its generative capabilities. To ensure an unbiased representation, the generation prompts were chosen at random, and the resulting visualizations are presented without curation or cherry-picking.

### E.5 FAKE TRAJECTORY VISUALIZATION

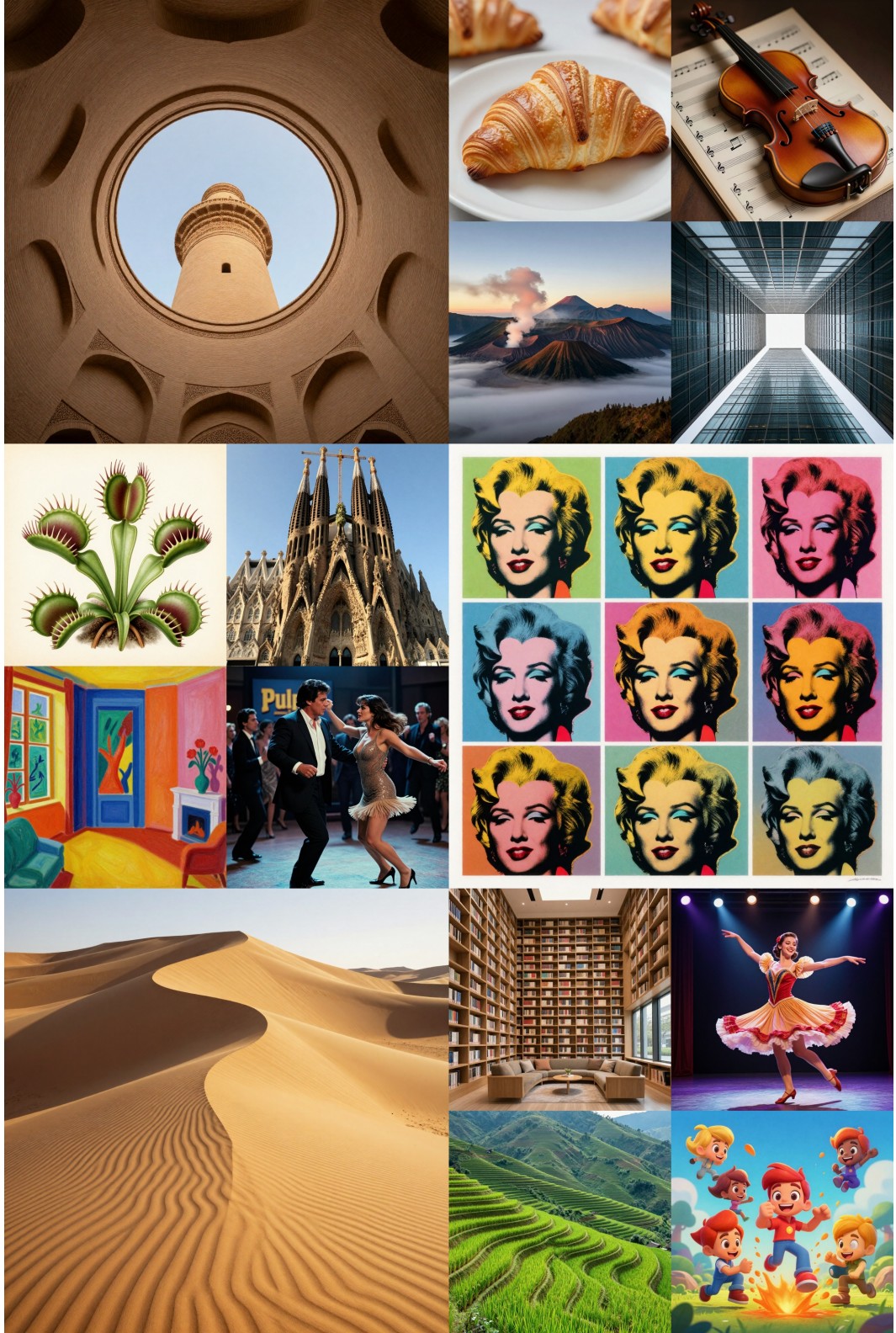

Figure 7: **Visualization of Qwen-Image-TWINFLOW (NFE=4).** Each image is of 1328×1328 resolution.

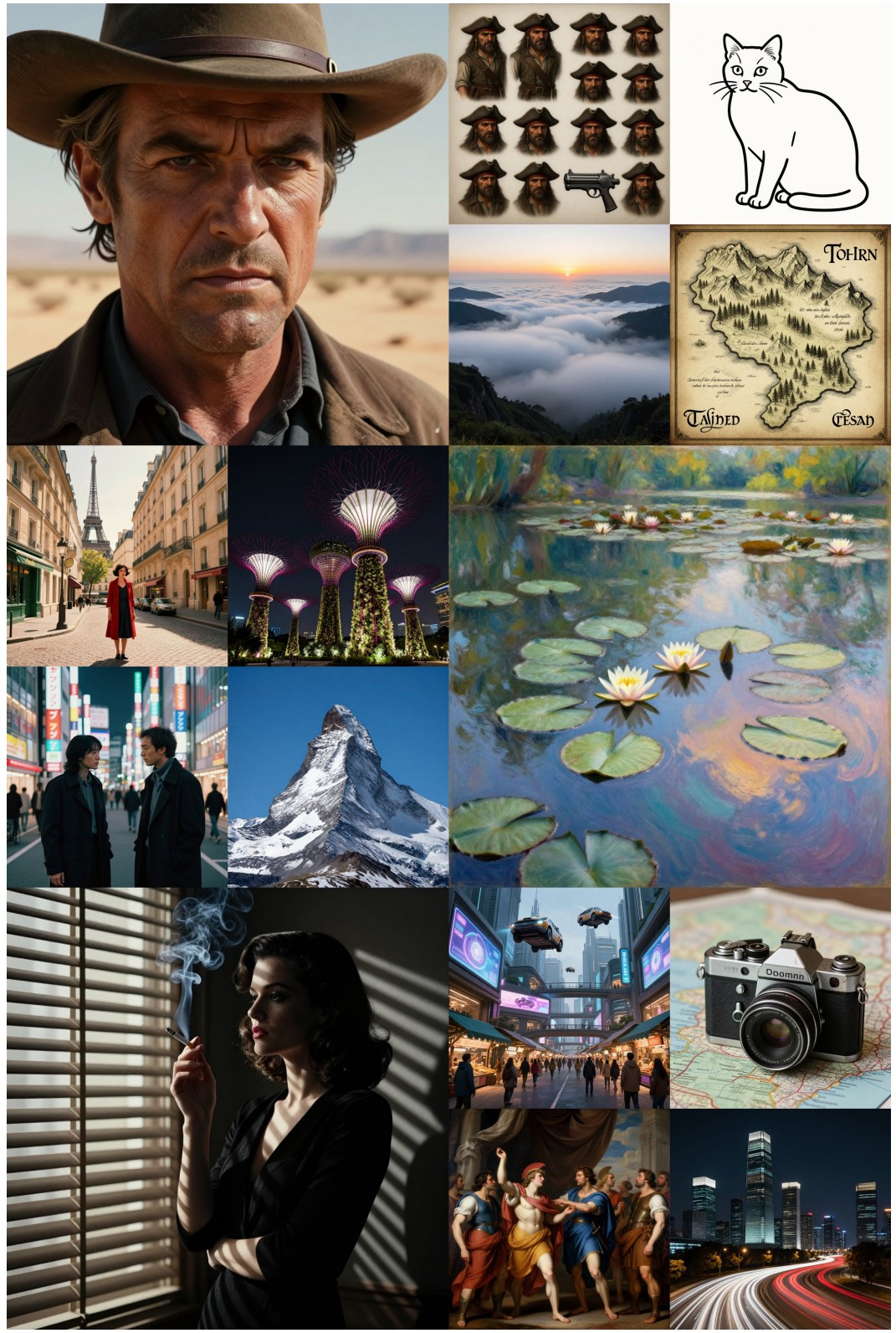

Figure 8: **Visualization of Qwen-Image-TWINFLOW (NFE=4).** Each image is of 1328×1328 resolution.

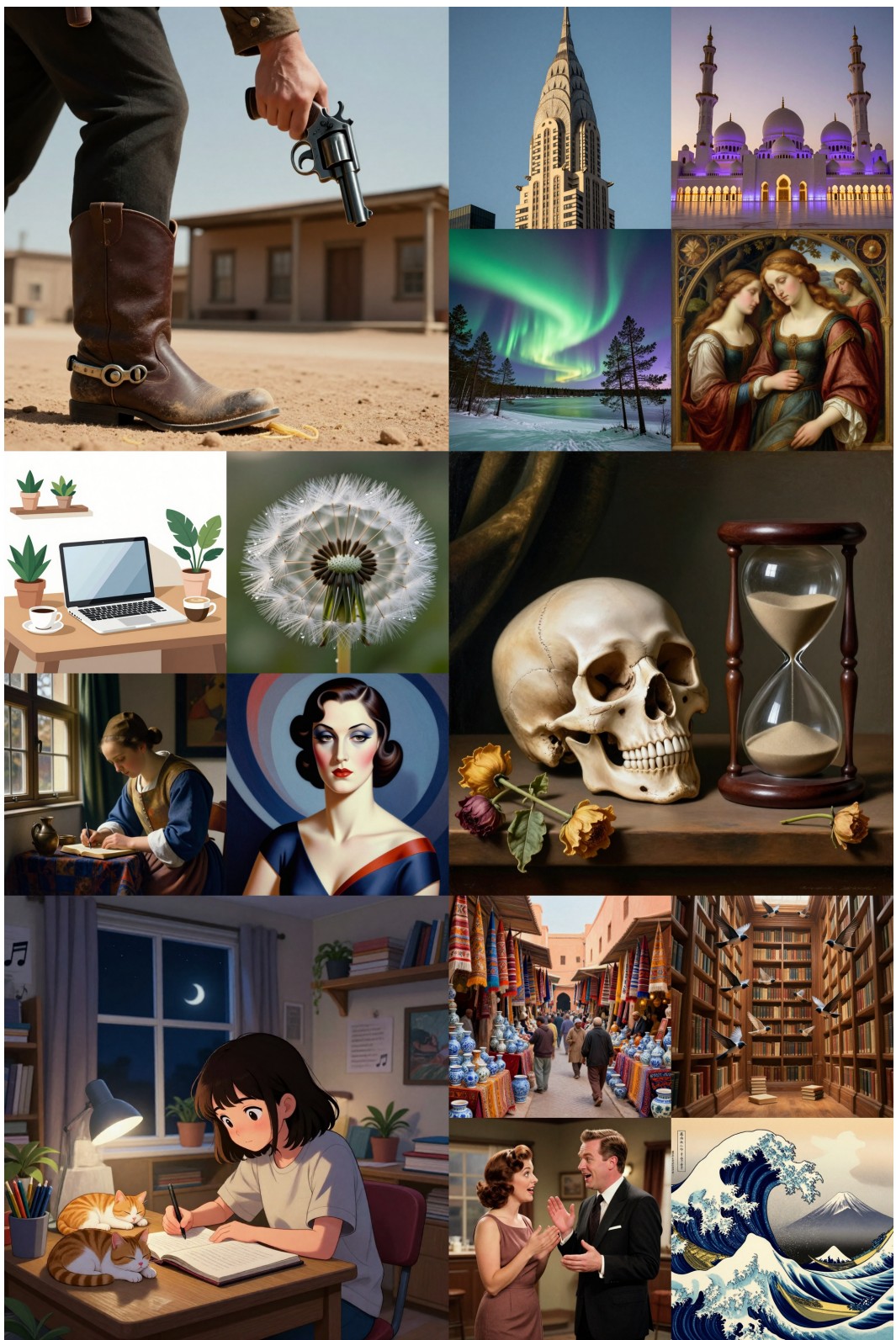

Figure 9: **Visualization of Qwen-Image-TWINFLOW (NFE=4).** Each image is of 1328×1328 resolution.

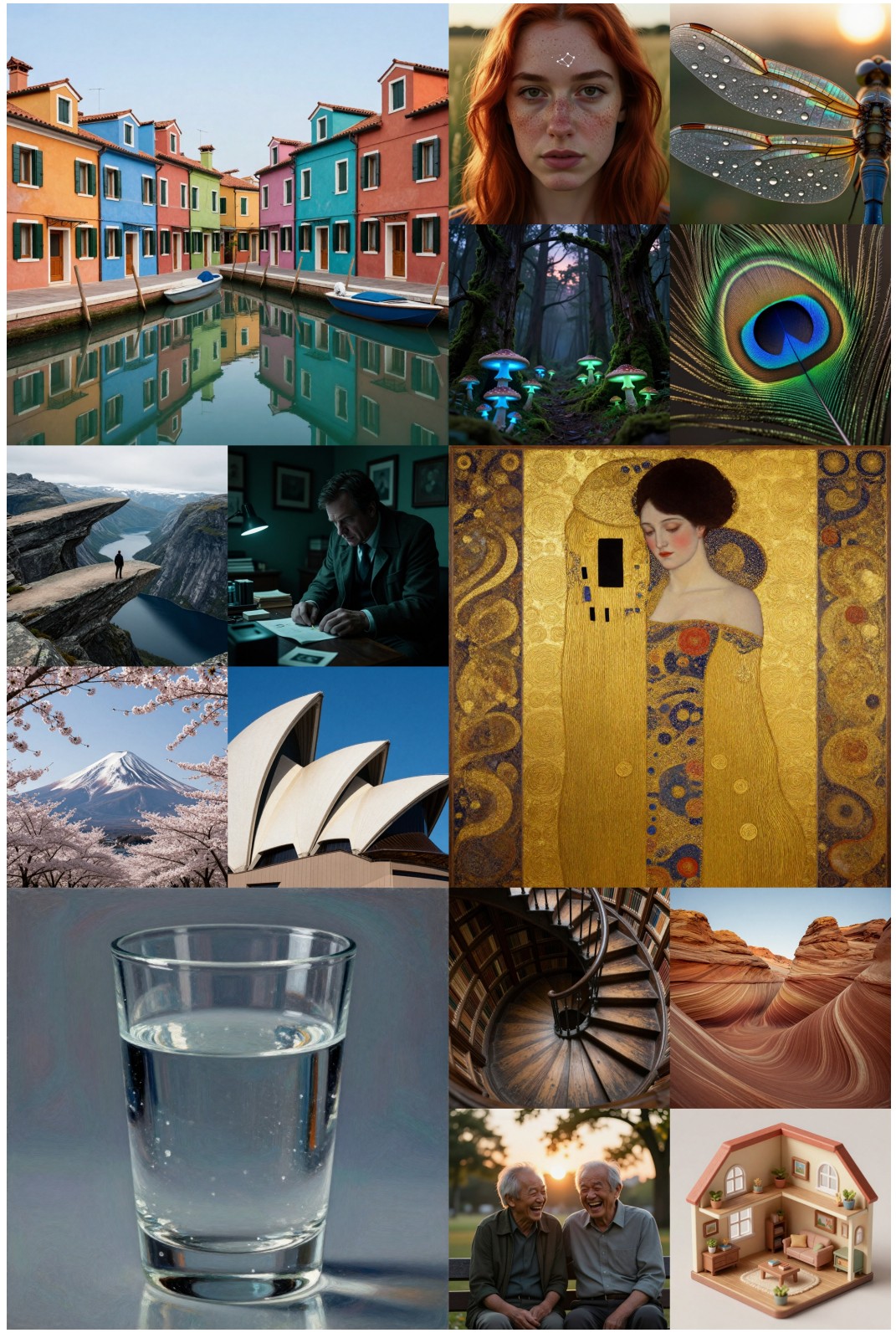

Figure 10: **Visualization of Qwen-Image-TWINFLOW (NFE=4).** Each image is of 1328×1328 resolution.

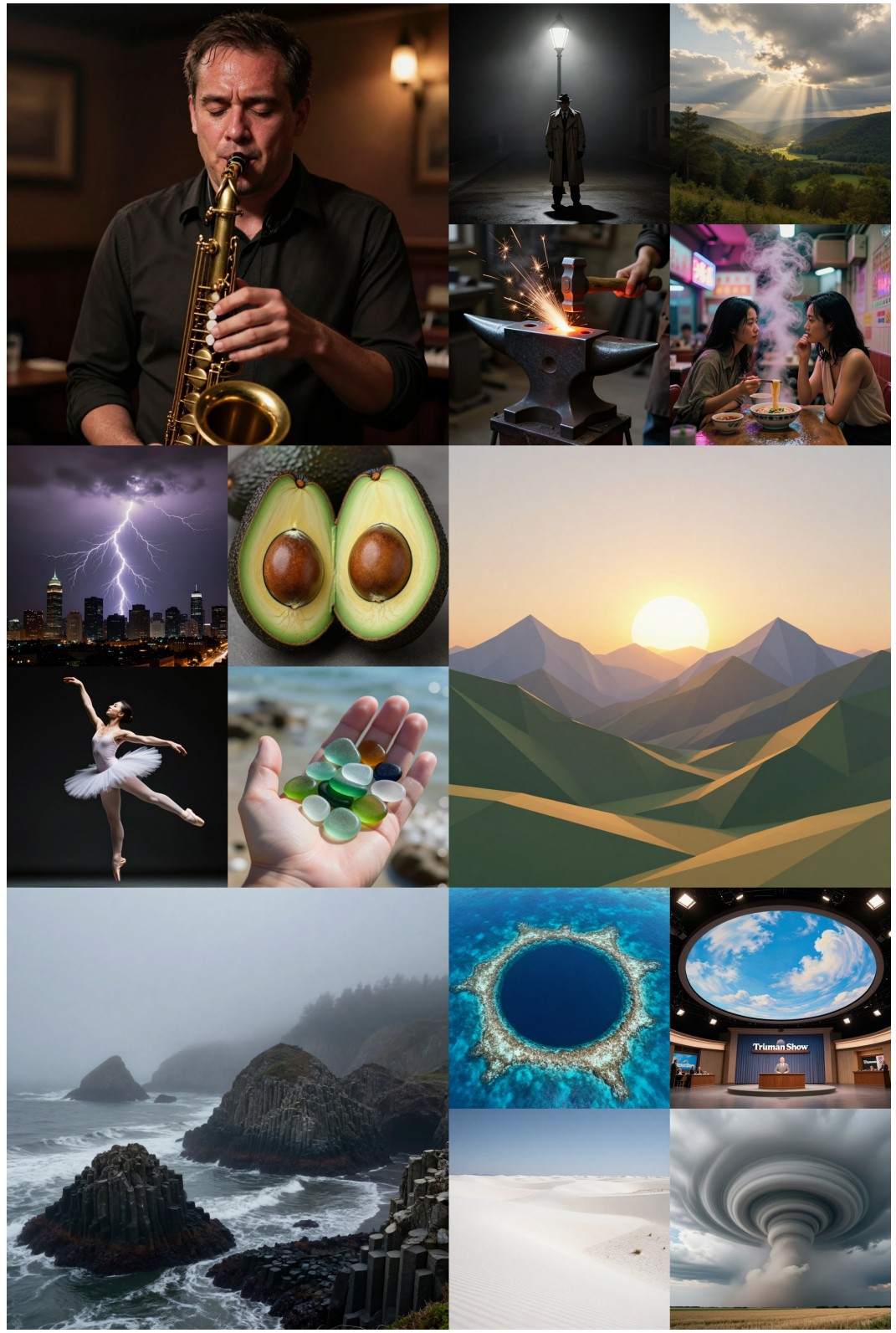

Figure 11: **Visualization of Qwen-Image-TWINFLOW (NFE=4).** Each image is of 1328×1328 resolution.

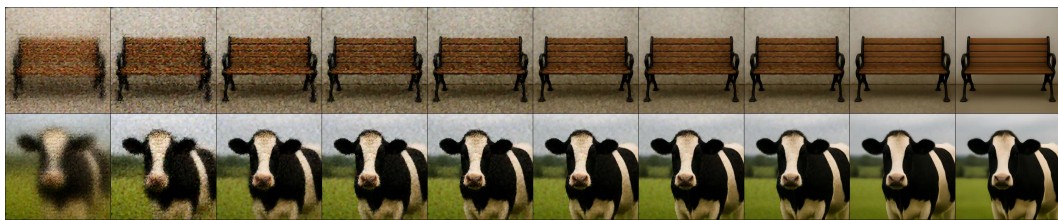

Figure 12: **Visualization of the fake trajectory (NFE=20).** The fake images have significant visual difference comparing to real images.

