# OpenReview forum: "TwinFlow: Realizing One-step Generation on Large Models with Self-adversarial Flows"
_ICLR.cc/2026/Conference — ICLR 2026 Poster_

### Official Review · Reviewer_qqUP · 2025-10-26

**Soundness:** 3
**Presentation:** 3
**Contribution:** 3
**Rating:** 6
**Confidence:** 5

**Summary:**

This paper proposes a new one-step generation method named TwinFlow. The method applies a self-adaversarial manner: First, it extend the common $[0,1]$ time interval into $[-1,1]$. Then it uses the $[0,1]$ and $[-1,0]$ interval for training a true and fake diffusion model, respectively. And a loss is used to minimize the difference of the true and fake velocities. Experiments show that the proposed method achieves strong performance on text-to-image tasks, and also verify its scalability on a 20B model.

**Strengths:**

- I like its feature where only one model is trained, and that it achieves good performance on T2I models without GAN. Compared to SiD and VSD, it combines the pretrained teacher and the true score net, while makes the fake score net online, which is a good improvement.
- The performance is strong, especially for the scalabity on large-scale (20B) models.

**Weaknesses:**

- This method is similar to existing distillation methods like SiD, DMD and VSD, for example, DMD requires a trained diffusion teacher and a online-training generator, corresponding to the $t\in [0,1]$ part of TwinFlow; the auxiliary fake score network corresponds to the $t\in[-1,0]$ part. The authors should provide a detailed comparison in their paper.
- Since this approach integrate the function of the three models in DMD into one model, there is potential performance degration since the total model capacity is limited.
- Though with better performance, the forward and backward pass of TwinFlow seems much more complicated than consistency training, the authors are expected to provide the theoretical and practical training cost comparison (also see questions).

**Questions:**

- How do you generate $\\boldsymbol{x}^{\text{fake}}$? I suppose $\\boldsymbol{x}^{\text{fake}}=\\boldsymbol{x}\_t+(0-t)\\boldsymbol{F}_{\theta}(\\boldsymbol{x}\_t,t,0)$ under the OT-FM framework. If so, is the sample generated by the diffusion ODE idential to that generated by the generator trained wiith DMD loss (rectify loss in the paper) in particle or just in distribution? How do this loss boost the 2 or more-step performance?
- In DMD and VSD, the training is alternatively (fake score net and generator). Why can your method be free from alternate training?
- I found that the authors did not discuss the usage of CFG in the method, and I also notice that in Figure 3, the TwinFlow results are without CFG. I wonder if CFG is applicable in TwinFlow, or can CFG improve the performance further if applicable?
- Is the DiT model on ImageNet trained from scratch or with teacher weights loaded? And, what is the 1-step performance? Also, are the results with or without CFG? If the $2.05$ FID is arrived without CFG, that is impressive.
- While there are multiple scores reported for 1-step generation, there are very few 1-step visualizations. Can you provide some (it won't affect my evaluation)?
- In consistency training/finetuning, the additional training cost compared to diffusion models is one forward pass. In your method, the extra cost seems much more. Can you elabarate the extra cost, and compare the practical memory and speed with consistency training (e.g., on ImageNet)?
- The $\lambda$ parameter only adjust the effect of $\mathcal{L}_\text{TwinFlow}$; do you also need to adjust the effect between adv loss and rectify loss?
- Is there a missing "-" in Eq. (6)?

---

> ### Author Response · Authors · 2025-11-26
> **Reply window of W1**
>
> > This method is similar to existing distillation methods like SiD, DMD and VSD, for example, DMD requires a trained diffusion teacher and a online-training generator, corresponding to the part of TwinFlow; the auxiliary fake score network corresponds to the part. The authors should provide a detailed comparison in their paper.
>
> We thank the reviewer for this comment. To thoroughly evaluate the advantages of our method on large-scale models, we conduct a comparative study on Qwen-Image-20B, benchmarking against SiD, DMD, VSD, and our proposed approach. The detailed comparison results are also shown in the Table 9 in the revised PDF.
>
> **The results show that our method can surpass VSD, DMD, and SiD on all benchmarks**, and the advantages of the one-model design become even more pronounced in the 20B full-parameter training setting.
>
> Besides numerical results, **we also find that there also exists mode collapse on DMD and SiD, which is the same as the phenomenon on Qwen-Image-Lightning.**
>
> | **Full Param Training (20B)**   | **NFE** | **GenEval** | **DPG-Bench** | **WISE** |
> | ------------------------------- | ------- | ----------- | ------------- | -------- |
> | **VSD (raw)**                   | N/A     | OOM         | OOM           | OOM      |
> | **DMD (raw)**                   | N/A     | OOM         | OOM           | OOM      |
> | **SiD (raw)**                   | N/A     | OOM         | OOM           | OOM      |
> | **VSD**                         | 1       | 0.67        | 84.44         | 0.22     |
> | **VSD**                         | 2       | 0.73        | 86.16         | 0.34     |
> | **DMD**                         | 1       | 0.81        | 84.31         | 0.47     |
> | **DMD**                         | 2       | 0.80        | 84.08         | 0.46     |
> | **SiD**                         | 1       | 0.77        | 87.05         | 0.42     |
> | **SiD**                         | 2       | 0.78        | 86.94         | 0.41     |
> | **RCGM (N=1)**                  | 1       | 0.48        | 73.78         | 0.21     |
> | **RCGM (N=1)**                  | 2       | 0.69        | 83.41         | 0.41     |
> | **RCGM (N=2)**                  | 1       | 0.56        | 76.15         | 0.31     |
> | **RCGM (N=2)**                  | 2       | 0.78        | 85.01         | 0.50     |
> | **RCGM (N=3)**                  | 1       | 0.52        | 74.80         | 0.27     |
> | **RCGM (N=3)**                  | 2       | 0.75        | 84.18         | 0.48     |
> | **Ours (N=1)**                  | 1       | 0.79        | 84.50         | 0.43     |
> | **Ours (N=1)**                  | 2       | 0.82        | 85.29         | 0.49     |
> | **Ours (N=2)**                  | 1       | 0.85        | 85.44         | 0.51     |
> | **Ours (N=2)**                  | 2       | 0.86        | 86.35         | 0.55     |
> | **Ours (N=3)**                  | 1       | 0.88        | 85.58         | 0.54     |
> | **Ours (N=3)**                  | 2       | 0.89        | 86.74         | 0.56     |
> | **Ours (N=2, longer training)** | 1       | 0.89        | 87.54         | 0.57     |
> | **Ours (N=2, longer training)** | 2       | 0.90        | 87.80         | 0.59     |

---

> ### Author Response · Authors · 2025-11-26
> **Reply window of W2**
>
> > Since this approach integrate the function of the three models in DMD into one model, there is potential performance degration since the total model capacity is limited.
>
> We thank the reviewer for this comment. We argue that the parameter budget of LoRA does not significantly constrain model performance. To support this claim, we provide additional experiments with LoRA rank of 32 and 128, as well as results from full-parameter training, demonstrating that performance remains robust across different adaptation configurations.
>
> |       | NFE  | GenEval | DPG-Bench | WISE |
> | ----- | ---- | ------- | --------- | ---- |
> | r=32  | 1    | 0.84    | 85.24     | 0.51 |
> | r=64  | 1    | 0.86    | 86.52     | 0.54 |
> | r=128 | 1    | 0.86    | 86.84     | 0.54 |
> | r=32  | 2    | 0.85    | 86.48     | 0.55 |
> | r=64  | 2    | 0.87    | 87.64     | 0.57 |
> | r=128 | 2    | 0.87    | 87.72     | 0.57 |

---

> ### Author Response · Authors · 2025-11-26
> **Reply window of W3 & Q6**
>
> > Though with better performance, the forward and backward pass of TwinFlow seems much more complicated than consistency training, the authors are expected to provide the theoretical and practical training cost comparison (also see questions).
>
> > In consistency training/finetuning, the additional training cost compared to diffusion models is one forward pass. In your method, the extra cost seems much more. Can you elabarate the extra cost, and compare the practical memory and speed with consistency training (e.g., on ImageNet)?
>
> We thank the reviewer for this comment. We conduct comparisons in the theoretically minimal setting, without employing CFG augmentation or other performance-enhancing techniques. For any training step, we denote:
>
> - batch size as `b`,
> - a forward pass with gradient computation as `fwd (with grad)` ,
> - a forward pass without gradient computation as `fwd (no grad)` ,
> - a Jacobian-vector product (JVP) as `jvp`.
>
> **For standard sCM or MeanFlow training, there includes:**
>
> - one `jvp`  computation of of batch size `b` .
>
> **For our TwinFlow training, there includes:**
>
> - one `fwd (with grad)`  of batch size `b` ,
> - one or two `fwd (no grad)`  of base loss of batch size `b` ,
> - one `fwd (with grad)`  of adv loss of  batch size `b/3` ,
> - two `fwd (no grad)`  of rectify loss of  batch size `b/3` .
>
> The factor of $\frac{1}{3}$ here arises because we use $\lambda = \frac{1}{3}$. The training speed and memory usage is averaged in 100 steps.
>
> The results show that the computational cost of JVP is extremely high—so much so that it forces the training batch size to be reduced and slows down training by approximately 2× (e.g., at batch size 128). Although our method involves a few additional forward passes, **its overall overhead is significantly lower than that of JVP.**
>
> |              | **Global Batch Size** | **Training Step / Second $\uparrow$** | **Memory (GB)** |
> | ------------ | --------------------- | ------------------------------------- | --------------- |
> | **sCM**      | 512                   | -                                     | OOM             |
> | **MeanFlow** | 512                   | -                                     | OOM             |
> | **TwinFlow** | 512                   | 3.40                                  | 438             |
> | **sCM**      | 256                   | -                                     | OOM             |
> | **MeanFlow** | 256                   | -                                     | OOM             |
> | **TwinFlow** | 256                   | 4.74                                  | 377             |
> | **sCM**      | 128                   | 2.45                                  | 515             |
> | **MeanFlow** | 128                   | 2.45                                  | 515             |
> | **TwinFlow** | 128                   | 5.67                                  | 305             |

---

> ### Author Response · Authors · 2025-11-26
> **Reply window of Q1**
>
> > How do you generate $x^{\mathrm{fake}}$ ? I suppose $x^{\mathrm{fake}}=x_t+(0-t) F_{\theta}(x_t,t,0)$ under the OT-FM framework.
>
> We thank the reviewer for this insightful comment. The understanding is correct, the $x^{\mathrm{fake}}$ is generated by $x^{\mathrm{fake}}=x_t+(0-t) F_{\theta}(x_t,t,0)$.
>
> > If so, is the sample generated by the diffusion ODE idential to that generated by the generator trained wiith DMD loss (rectify loss in the paper) in particle or just in distribution?
>
> If generator receives the same noise, the sample is matched in particle, otherwise in distribution.
>
> > How do this loss boost the 2 or more-step performance?
>
> Since we construct $x^{\mathrm{fake}}=x_t+(0-t) F_{\theta}(x_t,t,0)$, the samples generated by the diffusion ODE are directed from any point along the trajectory toward the endpoint. Consequently, the rectify loss can improve 2 or more-step performance, with empirical gains being most pronounced in the 2–4 steps regime.

---

> ### Author Response · Authors · 2025-11-26
> **Reply window of Q2**
>
> > In DMD and VSD, the training is alternatively (fake score net and generator). Why can your method be free from alternate training?
>
> We thank the reviewer for this comment. We clarify that the one-model feature of the TwinFlow allows us to embed the computation of the two additional losses within a single training step. For implementation simplicity, we compute the TwinFlow loss on a subset of samples within each batch, thereby integrating it seamlessly into one unified training step and avoiding the need for alternating between fake score and generator.

---

> ### Author Response · Authors · 2025-11-26
> **Reply window of Q3**
>
> > I found that the authors did not discuss the usage of CFG in the method, and I also notice that in Figure 3, the TwinFlow results are without CFG. I wonder if CFG is applicable in TwinFlow, or can CFG improve the performance further if applicable?
>
> We thank the reviewer for this comment. We clarify that:
>
> - CFG augmentation was used during training. All sampling results and visualizations reported in the submission were generated without CFG.
> - CFG remains applicable to TwinFlow. Under few-step sampling, applying CFG tends to cause over-saturation or over-contrasting in generated images. Under multi-step sampling, CFG can still improve performance.

---

> ### Author Response · Authors · 2025-11-26
> **Reply window of Q4**
>
> > Is the DiT model on ImageNet trained from scratch or with teacher weights loaded? And, what is the 1-step performance? Also, are the results with or without CFG? If the FID is arrived without CFG, that is impressive.
>
> We thank the reviewer for this comment. We clarify that: 1) The DiT model on ImageNet are trained from scratch. 2) The 1-NFE FID is 3.17. 2) All evaluations were obtained without using CFG.

---

> ### Author Response · Authors · 2025-11-26
> **Reply window of Q5**
>
> > While there are multiple scores reported for 1-step generation, there are very few 1-step visualizations. Can you provide some (it won't affect my evaluation)?
>
> We thank the reviewer for this comment. We have updated 1-NFE visualizations in the appendix Fig. 12 of the revised PDF.

---

> ### Author Response · Authors · 2025-11-26
> **Reply window of Q7 & Q8**
>
> > The $\lambda$ parameter only adjust the effect of $\mathcal{L}_{\mathrm{TwinFlow}}$; do you also need to adjust the effect between adv loss and rectify loss?
>
> We thank the reviewer for this comment. The coefficients of adv loss and the rectify loss are both 1, without any additional adjustment.
>
> > Is there a missing "-" in Eq. (6)?
>
> We thank the reviewer for pointing out and apologize for the typo in the submission. There is a missing “-” in Eq. 6. We will make revisions in the PDF (revisions will be highlighted in green).

---

> > ### Comment · Reviewer_qqUP · 2025-11-27
> >
> > Thank you for the comprehensive feedback. I still have questions as follows: i) As is known, we must train DMD alternatively; while in your loss functions, they are like DMD but are trained without alternative optimization. If I am correct, your base loss contains both standard FM loss and consistency loss. It is the introduction of consistency loss that allows to eliminate alternative training. Have you tried if your losses still work without consistency loss (that is, using only FM loss in the base loss)? ii) The comparison to sCM/MeanFlow seems unfair on the jvp operation. The differentiation in CMs can be realized with analytic jvp or approximate finite difference, where the latter should not deviate much compared with torch.jvp. This fact is also verified in your approach, where you also utilize consistency loss in its finite-difference version. iii) "The samples generated by the diffusion ODE are directed from any point along the trajectory toward the endpoint. Consequently, the rectify loss can improve 2 or more-step performance, with empirical gains being most pronounced in the 2–4 steps regime."---Do you mean that there is no explicit supervision for 2- or more-step samples, but they can be naturally improved by only the one-step supervision?

---

> > > ### Author Response · Authors · 2025-11-27
> > > **Reply window of further questions**
> > >
> > > We thank the reviewer for timely feedback.
> > >
> > > - **Clarification on base loss and experiment of only using FM loss.**  We clarify that the base loss is a any-step loss (c.f. Eq. 1), which is not a direct combination of FM loss and CM loss. We add an experiment of using only FM loss, the results show that using only FM loss has some performance drop but can still work.
> > >
> > > |  | NFE | GenEval | DPG-Bench | WISE |
> > > | --- | --- | --- | --- | --- |
> > > | TwinFlow + base loss | 1 | 0.85 | 85.44 | 0.51 |
> > > | TwinFlow + base loss | 2 | 0.86 | 86.35 | 0.55 |
> > > | TwinFlow + FM loss | 1 | 0.78 | 83.82 | 0.48 |
> > > | TwinFlow + FM loss | 2 | 0.80 | 84.02 | 0.51 |
> > >
> > > Additionally, we also did a test on MNIST using Maximum Mean Discrepancy (MMD) [1] as the metric. The results show that using only FM loss can work well. Corresponding 1-NFE visualizations are provided in the revised PDF appendix Fig. 16.
> > >
> > > |  | RCGM | TwinFlow + base loss | TwinFlow + FM loss |
> > > | --- | --- | --- | --- |
> > > | MMD $\downarrow$ | 0.02063 | 0.01762 | 0.01963 |
> > > - **Re-comparison with sCM/MeanFlow training.** Thanks for the advice. We implement the JVP operation by finite-difference in sCM and MeanFlow and compare the speed and memory usage. Our method is slightly slower and increases a few memory usage as we compute additional $L_{TwinFlow}$ on $\frac{1}{3}$ of the batch. However, **this marginal cost is well-justified by the improvements in few-step generation quality, as shown in the following table (same setup in previous Qwen-Image-20B full parameter training).**
> > >
> > > |  | **Global Batch Size** | **Training Step / Second $\uparrow$** | **Memory (GB)** |
> > > | --- | --- | --- | --- |
> > > | **sCM** | 512 | 4.44 | 421 |
> > > | **MeanFlow** | 512 | 4.44 | 421 |
> > > | **TwinFlow** | 512 | 3.40 | 438 |
> > > | **sCM** | 256 | 6.78 | 302 |
> > > | **MeanFlow** | 256 | 6.78 | 302 |
> > > | **TwinFlow** | 256 | 4.74 | 377 |
> > > | **sCM** | 128 | 7.99 | 226 |
> > > | **MeanFlow** | 128 | 7.99 | 226 |
> > > | **TwinFlow** | 128 | 5.67 | 305 |
> > >
> > > |  | NFE $\downarrow$ | GenEval | DPG-Bench | WISE |
> > > | --- | --- | --- | --- | --- |
> > > | MeanFlow (JVP-free) | 4 | 0.44 | 83.28 | 0.34 |
> > > | MeanFlow (JVP-free) | 8 | 0.49 | 83.81 | 0.37 |
> > > | sCM (JVP-free) | 4 | 0.62 | 85.37 | 0.44 |
> > > | sCM (JVP-free) | 8 | 0.60 | 85.54 | 0.45 |
> > > | Ours | 1 | 0.85 | 85.44 | 0.51 |
> > > | Ours | 2 | 0.86 | 86.35 | 0.55 |
> > > - **Explanation on rectify loss.** The rectify loss does not explicitly supervise 2 or more-step generation. We speculate that it can still enhance performance in the 2–4 step regime to some extent. The fact that TwinFlow outperforms RCGM at 2 steps supports this conjecture.
> > >
> > > [1] Gretton, A., Borgwardt, K., Rasch, M., Schölkopf, B., & Smola, A. (2006). A kernel method for the two-sample-problem. *Advances in neural information processing systems*, *19*.

---

> > > > ### Comment · Reviewer_qqUP · 2025-11-28
> > > >
> > > > The authors' rebuttal addressed my questions. The empirical results that 'TwinFlow+FM loss only' still works remains interesting to me. I've decided to keep my positive rating. Thank you.

---

> ### Author Response · Authors · 2025-11-28
>
> Thanks for the reviewer's constructive feedback, we will incorporate rebuttal experiments into the PDF.

---

### Official Review · Reviewer_ESMU · 2025-10-29

**Soundness:** 2
**Presentation:** 2
**Contribution:** 2
**Rating:** 6
**Confidence:** 4

**Summary:**

TwinSteps is a framework that aims to reduce the number of inference steps in diffusion / flow matching models to as few as 1 step. It bypasses the need for distillation and avoids standard adversarial training (e.g. GAN style). The approach strongly builds on the RCGM method, but adds three main additional components: extending the time domain from [0,1] to [-1,1] and adding an adversarial as well as a rectification loss. On text-to-image tasks, the method outperforms multiple baselines and is shown to be scalable by applying it on the largest open-source multi-modal model, Qwen-Image-20B.

**Strengths:**

- While many distillation or few step approaches leverage additional trained or teacher networks, this method does not require any additional separate network.
- While strongly leveraging ideas by the DMD (Distribution Matching Distillation paper) paper, the authors found an elegant way to implement the idea of an adversarial network component into the existing flow matching / diffusion frameworks by extending the time domain to [-1,1] where [-1,0] corresponds to the fake data domain.
- The presented method shows strong results on the image generation benchmarks GenEval, DPG-Bench and WISE

**Weaknesses:**

- Intuition and explanation:
While the intuition of an adversarial component is clear in DMD is clear, certain design choices had to be taken to make this method work:
a) Representing fake data by negative time steps
b) Having the adversarial loss learning the negative trajectory
c) A rectification loss that should straighten the trajectory
Further, during training N=2 is adopted, meaning 2nd order approximations are done in the RCGM framework, giving a strong few-step prior.
The paper does not fully explain why the negative time trajectory is required and why the rectification component ultimately converges to a single step solution. If this was not investigated but only empirically observed, this needs to be clearly pointed out
The strength of the method could be better explained in more simple settings such as CIFAR10 or Image1K.

- Evaluation and Results:
Almost all evaluations have been done on text-to-image benchmarks. This comes with additional complexity in the evaluation as e.g. diversity of images for the same text prompt multiple times (as pointed out by the authors in Table 2). While not explicitly mentioned in the main paper, the method seems to perform worse on other (simpler) benchmarks as e.g. Image-1K 256x256. In fact the 2 step FID of Twinflow trained for 801 epochs mentioned in the appendix (FID=2.05) is larger than the pure RCGM FID value after 424 epochs (FID=1.92, see https://github.com/LINs-lab/RCGM/blob/main/assets/paper.pdf).
Qwen-Image-Lightning is 1 step leader on the DPG benchmark and should be marked like this in Table 2

- Distillation / Fine Tuning vs. Full training method:
Qwen-Image-TwinFlow (and possibly also TwinFlow-0.6B and TwinFlow-1.6B, see question below) leverages a pretrained model that is fine-tuned. In this case the pretrained model somehow acts as a “teacher” adding stability to the fine-tuning i.e. as the fake data predictions.
However, when training from scratch the optimization problem becomes significantly harder. Possible stability issues and their solutions are not discussed in the paper.

- Unclear writing:
In several parts of the paper, i.e. the methodology section, it is not always easy to follow the equations as variable names are not always defined or used in a consistent manner (see questions below)

**Questions:**

- In Eq. 1 it is not defined what $f_\Theta-$ is?
- In Eq. 2, does $z^{fake}$ correspond to $z$ that is also used in the base component of the loss to ensure alignment?
- Eq. 2 has the objective to learn the negative time time trajectory. Why is it chosen differently to the normal base loss?
- In Eq. 8 and 9: Did I understand correctly that $x_t$ corresponds to $x_{t’}^{fake}$?
- On intuition and explanation of weaknesses: Can you give more information on the addressed point?
- Instead of using a negative time trajectory, would a simple boolean condition as model input that indicates fake or real data also work?
- Given the adversarial components do help the 1 / 2 step predictions, do they also help to improve many step predictions e.g. 50 or 100 NFE’s ?
- On the Qwen-Image-Lightning diversity discussion: The shown results in the appendix support the “almost identical image” claim, but do - not seem sufficient. Can you quantify or provide more examples to support this claim?
- In Table 2: Why is the performance increase of the method better on Qwen-Image than on OpenUni for 1 and 2 steps?
- Can you clarify if TwinFlow-0.6B and TwinFlow-1.6B leverage a pretrained backbone that is full parameter finetuned or if these two models are trained ab-initio / from scratch? What about the ImageNet-1K results mentioned in the appendix?

- Minor comment:
Typo in Line 201: “to match with us”, probably “to match each other”?

---

> ### Author Response · Authors · 2025-11-26
> **Reply window of W1 & Q5**
>
> > Intuition and explanation: While the intuition of an adversarial component is clear in DMD is clear, certain design choices had to be taken to make this method work: a) Representing fake data by negative time steps b) Having the adversarial loss learning the negative trajectory c) A rectification loss that should straighten the trajectory Further, during training N=2 is adopted, meaning 2nd order approximations are done in the RCGM framework, giving a strong few-step prior. The paper does not fully explain why the negative time trajectory is required and why the rectification component ultimately converges to a single step solution. If this was not investigated but only empirically observed, this needs to be clearly pointed out The strength of the method could be better explained in more simple settings such as CIFAR10 or Image1K.
>
> > On intuition and explanation of weaknesses: Can you give more information on the addressed point?
>
> We thank the reviewer for this insightful comment. We clarify the following points:
>
> - **2-order approximation in the RCGM is not a strong few-step prior:** This choice was made solely to ensure a fair comparison with the original RCGM setup. To address potential concerns that higher-order RCGM (N ≥ 2) might implicitly encode a few-step bias, we have added supplementary experiments using RCGM with N = 1, 2, 3 under identical conditions. These results show that increasing the approximation order does not lead to significant improvements in 1-step or 2-step generation performance. The detailed comparison results are also shown in the Table 9 in the revised PDF.
>
> | **Full Param Training (20B)**   | **NFE** | **GenEval** | **DPG-Bench** | **WISE** |
> | ------------------------------- | ------- | ----------- | ------------- | -------- |
> | **RCGM (N=1)**                  | 1       | 0.48        | 73.78         | 0.21     |
> | **RCGM (N=1)**                  | 2       | 0.69        | 83.41         | 0.41     |
> | **RCGM (N=2)**                  | 1       | 0.56        | 76.15         | 0.31     |
> | **RCGM (N=2)**                  | 2       | 0.78        | 85.01         | 0.50     |
> | **RCGM (N=3)**                  | 1       | 0.52        | 74.80         | 0.27     |
> | **RCGM (N=3)**                  | 2       | 0.75        | 84.18         | 0.48     |
> | **Ours (N=1)**                  | 1       | 0.79        | 84.50         | 0.43     |
> | **Ours (N=1)**                  | 2       | 0.82        | 85.29         | 0.49     |
> | **Ours (N=2)**                  | 1       | 0.85        | 85.44         | 0.51     |
> | **Ours (N=2)**                  | 2       | 0.86        | 86.35         | 0.55     |
> | **Ours (N=3)**                  | 1       | 0.88        | 85.58         | 0.54     |
> | **Ours (N=3)**                  | 2       | 0.89        | 86.74         | 0.56     |
> | **Ours (N=2, longer training)** | 1       | 0.89        | 87.54         | 0.57     |
> | **Ours (N=2, longer training)** | 2       | 0.90        | 87.80         | 0.59     |
>
> - **On the use of negative time trajectories**: In our early experiments, we explored alternative formulations for the self-adversarial component, including training on a positive time interval such as $[1, 2]$ (mapping $t=1$ to noise and $t=2$ to data). This experiment resulted in a significant degradation of performance compared to the $[-1, 0]$ interval. We attribute this to using $[1, 2]$ interval were insufficient to produce a robust contrastive signal. This "easier" trajectory may fail to generate "hard" fake samples, thereby providing inadequate contrast for the model to learn clear differences between real and fake data.
> - **On the rectification component**: Inspired by previous distribution matching methods such as DMD, our rectification component minimizes the KL divergence, which effectively prioritizes matching the final data distribution over following a specific trajectory. Crucially, **by translating the score difference into a velocity correction (Eq. 7), the loss explicitly realigns the model's velocity field to point directly from the noise input towards the high-density data regions. This straighten the trajectory, ensuring that a single linear integration step is sufficient to bridge the gap between noise and data.**
> - **On the strength of our method:** we clarify that the benefits are primarily evident on large models. Our core contribution lies in the **“one-model” design**, which unifies the generator, fake score, and real score into a single model—**without requiring standard adversarial training**. This eliminates the substantial GPU memory overhead and avoids introducing additional trainable modules, thereby enabling straightforward scalability to large models.

---

> ### Author Response · Authors · 2025-11-26
> **Reply window of W2**
>
> > Evaluation and Results: Almost all evaluations have been done on text-to-image benchmarks. This comes with additional complexity in the evaluation as e.g. diversity of images for the same text prompt multiple times (as pointed out by the authors in Table 2). While not explicitly mentioned in the main paper, the method seems to perform worse on other (simpler) benchmarks as e.g. Image-1K 256x256. In fact the 2 step FID of Twinflow trained for 801 epochs mentioned in the appendix (FID=2.05) is larger than the pure RCGM FID value after 424 epochs (FID=1.92, see [https://github.com/LINs-lab/RCGM/blob/main/assets/paper.pdf](https://github.com/LINs-lab/RCGM/blob/main/assets/paper.pdf)).
>
> We thank the reviewer for this comment. We clarify that:
>
> - **The issue highlighted regarding Qwen-Image-Lightning in Table 2 is not intended to underscore the complexity of text-to-image evaluation**, but rather to point out a specific concern: its limited generation diversity may lead to biased evaluation results on benchmarks such as GenEval and DPG-Bench.
>   - Note that these benchmarks generate four images per prompt, and if the model outputs nearly identical images across all four samples, the computed metrics (e.g., based on averaged scores) can become misleadingly optimistic—effectively allowing the model to "game" the benchmark through low-variance, repetitive outputs rather than demonstrating genuine multi-modal generation capability.
> - We primarily evaluate on text-to-image generation to assess the effectiveness of our method on large-scale models and in practically relevant text-to-image settings, as prior few-step approaches have not been scaled to the 20B regime due to the high cost of maintaining the generator, real score, and fake score at the same time.
> - To analyze the problem of slightly inferior performance on ImageNet, First, we ruled out model capacity as a cause, given TwinFlow-0.6B's strong performance on T2I. Second, compared to T2I, the only difference on Imagenet is the need to enable dropout for learning unconditional generation ($L_{base, uncond}$), when combined with other losses ($L_{base, cond}$, $L_{TwinFlow}$) might lead to conflicts. In recent multimodal generative models (e.g. Qwen-Image), there is no need to enable dropout for learning unconditional generation, thus avoiding potential conflicts.
>
> We would like to further clarify that **our core contribution lies in the simplicity and scalability of the "one model" design.** To validate this, **we present 20B full-parameter training results,** demonstrating that our method's advantages are fully realized at scale where such conflicts are avoided. (See next window).
>
> > Qwen-Image-Lightning is 1 step leader on the DPG benchmark and should be marked like this in Table 2
> - We have updated the PDF to mark this.

---

> ### Author Response · Authors · 2025-11-26
> **Reply window of W3**
>
> > Distillation / Fine Tuning vs. Full training method: Qwen-Image-TwinFlow (and possibly also TwinFlow-0.6B and TwinFlow-1.6B, see question below) leverages a pretrained model that is fine-tuned. In this case the pretrained model somehow acts as a “teacher” adding stability to the fine-tuning i.e. as the fake data predictions. However, when training from scratch the optimization problem becomes significantly harder. Possible stability issues and their solutions are not discussed in the paper.
>
> We thank the reviewer for this comment.
>
> **Standard practice vs. our innovation:**
>
> In the realm of large-scale Text-to-Image (T2I) generation, training from scratch is prohibitively expensive. Therefore, it is standard practice to initialize few-step training from a pre-trained multi-step backbone [1,2,3].
>
> - **Traditional methods:** Approaches like Consistency Distillation or DMD/DMD2 explicitly require a separate, fixed pre-trained model to serve as a teacher network throughout training to provide supervision.
> - **Our approach:** A key innovation of our method is that **it does not require a fixed teacher model.** Instead, the "teacher" (multi-step) and "student" (few-step) components update simultaneously.
>
> **Addressing the “teacher” concern:**
>
> We understand the reviewer’s concern that when using LoRA, the frozen backbone might effectively act as a stabilizer or "teacher." To disprove this dependency and demonstrate stability, we conducted a **full-parameter training experiment on Qwen-Image-20B**.
>
> - In this setting, **all parameters are updated**, meaning no part of the model acts as a fixed teacher.
> - Despite the massive scale and the absence of frozen components, the training remained stable and converged effectively.
>
> |                                       | NFE         | GenEval  | DPG-Bench | WISE     |
> | ------------------------------------- | ----------- | -------- | --------- | -------- |
> | Qwen-Image                            | 50$\times$2 | 0.87     | 88.32     | 0.62     |
> | LoRA version (reported in submission) | 1           | 0.86     | 86.52     | 0.54     |
> | Full param training version           | 1           | **0.89** | **87.54** | **0.57** |
> | LoRA version (reported in submission) | 2           | 0.87     | 87.64     | 0.57     |
> | Full param training version           | 2           | **0.90** | **87.80** | **0.59** |
>
> [1] Yin, T., Gharbi, M., Zhang, R., Shechtman, E., Durand, F., Freeman, W. T., & Park, T. (2024). One-step diffusion with distribution matching distillation. In *Proceedings of the IEEE/CVF conference on computer vision and pattern recognition* (pp. 6613-6623).
>
> [2] Yin, T., Gharbi, M., Park, T., Zhang, R., Shechtman, E., Durand, F., & Freeman, B. (2024). Improved distribution matching distillation for fast image synthesis. *Advances in neural information processing systems*, *37*, 47455-47487.
>
> [3] Chen, J., Xue, S., Zhao, Y., Yu, J., Paul, S., Chen, J., ... & Xie, E. (2025). Sana-sprint: One-step diffusion with continuous-time consistency distillation. *arXiv preprint arXiv:2503.09641*.

---

> ### Author Response · Authors · 2025-11-26
> **Reply window of W4 & Q1,Q2,Q4**
>
> > Unclear writing: In several parts of the paper, i.e. the methodology section, it is not always easy to follow the equations as variable names are not always defined or used in a consistent manner (see questions below)
>
> > In Eq. 1 it is not defined what $f_{\theta-}$ is?
>
> > In Eq. 2, does $z^{fake}$ correspond to that is also used in the base component of the loss to ensure alignment?
>
> > In Eq. 8 and 9: Did I understand correctly that $x_t$ corresponds to $x_{t'}^{fake}$?
>
> We sincerely apologize for the confusion caused by the unclear notation and inadequate explanations. We provide a more detailed explanation of the meanings of these notations and make revisions in the PDF (revisions are highlighted in green).
>
> - $f_{\theta-}$ represents the model without gradient.
> - $z^{\mathrm{fake}}$ is a different noise, it does not need to be the same in the base component.
> - In Eq. 8, we instantiate the $x_t$ in Eq. 6 by $x_{t'}^{\mathrm{fake}}$.
> - In Eq. 9, $x_t$ is the exact perturbed version of the base component’s input, it is not $x_{t'}^{\mathrm{fake}}$.
>
> Our original intention was to present a general derivation in Eq. 6 and then substitute the actual $x_{t'}^{\mathrm{fake}}$ in Eq. 8. However, this has led to a notational overlap in Eq. 10, causing ambiguity.
>
> We have revised the notation in the PDF to eliminate this redundancy and ensure clarity. We thank the reviewer again for their careful corrections.

---

> ### Author Response · Authors · 2025-11-26
> **Reply window of Q3 & Q6**
>
> > Eq. 2 has the objective to learn the negative time time trajectory. Why is it chosen differently to the normal base loss?
>
> We thank the reviewer for this comment. We clarify that the model only needs to learn a standard multi-step flow matching objective on the negative time interval because it functions as a fake score. The base loss also learns few-step, which is not needed for fake score.
>
> > Instead of using a negative time trajectory, would a simple boolean condition as model input that indicates fake or real data also work?
>
> We thank the reviewer for this insightful idea. **Using a boolean condition to separate real and fake data could not work in practice because it can not effectively produce sufficient difference between real and fake data.**
>
> One of the most critical components of our method is the use of an additional negative time interval to enable the model to generate ‘fake’ samples that are sufficiently distinct from real samples. This distinction is essential: it ensures that the rectification part can produce a sufficiently pronounced difference, thereby providing an effective gradient signal.
>
> If instead, the input data are forcibly partitioned into real and fake categories solely based on a boolean conditioning signal (even though all samples are in fact real), the resulting distinction would be inadequate, leading to significant difficulties in optimization of the rectification part.
>
> **To further facilitate understanding, we provide visualizations of fake trajectories sampled from the negative time interval in the appendix Fig. 15 of the revised PDF.**

---

> ### Author Response · Authors · 2025-11-26
> **Reply window of Q7**
>
> > Given the adversarial components do help the 1 / 2 step predictions, do they also help to improve many step predictions e.g. 50 or 100 NFE’s ?
>
> We thank the reviewer for this comment. We provide more 50-NFE visualizations in the appendix Fig. 13 & 14. The 50-NFE visualizations have more details than 4 NFE ones (Fig. 10 & 11).

---

> ### Author Response · Authors · 2025-11-26
> **Reply window of Q8**
>
> > On the Qwen-Image-Lightning diversity discussion: The shown results in the appendix support the “almost identical image” claim, but do - not seem sufficient. Can you quantify or provide more examples to support this claim?
>
> We thank the reviewer for interests in the diversity discussion. [1] also points out the diversity problem and shows that DMD will cause diversity degradation in their Table 6.
>
> We quantify the similarity by using LPIPS metric. We perform the diversity evaluation on GenEval. Specifically, for each prompt, we compute the average pairwise LPIPS distance among the 4 generated samples. The final score is the mean of all scores across the entire samples. **The results quantify that** **DMD will cause obvious diversity degradation.**
>
> |                      | **NFE** | **Qwen-Image-Lightning** | **Ours** |
> | -------------------- | ------- | ------------------------ | -------- |
> | **LPIPS $\uparrow$** | 1       | 0.2996                   | 0.5044   |
> | **LPIPS $\uparrow$** | 2       | 0.3046                   | 0.5188   |
>
> [1] Jiang, D., Liu, D., Wang, Z., Wu, Q., Jin, X., Liu, D., ... & Yang, H. (2025). Distribution Matching Distillation Meets Reinforcement Learning. *arXiv preprint arXiv:2511.13649*.

---

> ### Author Response · Authors · 2025-11-26
> **Reply window of Q9**
>
> > In Table 2: Why is the performance increase of the method better on Qwen-Image than on OpenUni for 1 and 2 steps?
>
> We thank the reviewer for this insightful comment. The SANA-Sprint paper notes that when performing few-step distillation, the training process is highly prone to collapse unless QK-Norm is explicitly introduced. To address this, SANA-Sprint modified the model architecture prior to distillation by explicitly incorporating QK-Norm and subsequently retrained the modified model on its internal private dataset. **This indicates that the original SANA architecture suffers from inherent training instability under few-step scenarios.**
>
> OpenUni adopts SANA as the backbone of its DiT. However, since we do not have access to the internal training data used in SANA-Sprint and full retraining of the entire backbone network would entail prohibitive engineering and computational costs, we did not incorporate the QK-Norm architectural modification into the SANA variant employed by OpenUni. Consequently, under few-step training settings, **OpenUni inherits the instability of the original SANA architecture**, which limits its potential for improvement.
>
> In contrast, the Qwen-Image architecture already includes QK-Norm, which mitigate the instability from the architecture perspective, enabling our method to achieve more performance gains.

---

> ### Author Response · Authors · 2025-11-26
> **Reply window of Q10 & Q11**
>
> > Can you clarify if TwinFlow-0.6B and TwinFlow-1.6B leverage a pretrained backbone that is full parameter finetuned or if these two models are trained ab-initio / from scratch? What about the ImageNet-1K results mentioned in the appendix?
>
> We thank the reviewer for this comment. We clarify that TwinFlow-0.6B and TwinFlow-1.6B leverage a pretrained backbone of SANA-0.6B and SANA-1.6B, respectively. We use our method to do full parameter finetuning on them. The ImageNet-1K results in the appendix are trained from scratch.
>
> > Minor comment: Typo in Line 201: “to match with us”, probably “to match each other”?
>
> We thank the reviewer for pointing out and apologize for the typo in the submission. it should be “each other”. We will make revisions in the PDF (revisions will be highlighted in green).

---

> ### Author Response · Authors · 2025-11-28
> **Kind Reminder for Reviewer ESMU**
>
> Dear Reviewer ESMU,
>
> Thank you for your feedback and review of our paper. Your insights have been instrumental in helping us refine and improve our work. In response to the concerns you raised, **we have provided detailed explanations and new experiments to address each point**. Below, we summarize our key responses to your concerns:
>
> - **Justification of design choices:** We clarified that the 2nd-order approximation is not a strict requirement by providing new ablations ($N=1, 2, 3$) showing consistent performance. We also explained that the negative time trajectory is essential for generating "hard" fake samples to create robust contrast, whereas positive intervals or simple boolean conditions fail to provide sufficient gradients for rectification.
> - **Validation of “teacher” concern & stability:** To address concerns about reliance on a frozen "teacher," we conducted **full-parameter training on Qwen-Image-20B**. The results (GenEval 0.90) confirm that our method converges stably without a fixed backbone, validating the robustness and scalability of the "one-model" framework.
> - **Clarification on diversity & benchmarks:** We quantified the diversity collapse of Qwen-Image-Lightning using LPIPS (0.30 vs. our 0.50), demonstrating that our method preserves generation diversity while achieving high scores. We also clarified that performance variations on OpenUni stem from architectural instabilities (lack of QK-Norm) rather than limitations of our method.
> - **Revisions on notations:** We have revised the manuscript to clarify notations (e.g., $f_{\theta-}$, $z^{\mathrm{fake}}$), ensuring that the derivation is consistent and easy to follow.
>
> Once again, thank you for your time and effort in reviewing our paper. Your feedback has been invaluable in improving the quality of our manuscript.
>
> We believe we have fully addressed all of your concerns. Please do not hesitate to reach out if you have any further questions or comments.

---

### Official Review · Reviewer_xzEE · 2025-11-01

**Soundness:** 2
**Presentation:** 3
**Contribution:** 2
**Rating:** 4
**Confidence:** 4

**Summary:**

This paper introduces TWINFLOW, a novel framework for training one-step generative models that achieves efficient inference without requiring auxiliary discriminators or frozen teacher models. The key innovation is the "twin trajectory" concept, which extends the time interval from [0,1] to [-1,1], creating paired trajectories for real and fake data generation. By minimizing the velocity field differences between these trajectories, the method achieves self-adversarial training. The authors demonstrate TWINFLOW's effectiveness by successfully applying it to Qwen-Image-20B, achieving competitive 1-NFE performance that matches 100-NFE baselines while reducing computational costs by 100×.

**Strengths:**

- Successfully applying the method to 20B parameter models demonstrates promising scalability.
- The 1-NFE performance closely matching 100-NFE baselines is impressive.

**Weaknesses:**

- Theoretical flaws. If I understand correctly, the paper trains a model with a single-timestep condition to handle generation at arbitrary steps. **However, such a model is no longer a score model, and using it in reverse KL to compute the gradient of logp is theoretically flawed**.
- BLIP-3o-60K contains samples carefully generated using GenEval prompts, and training on BLIP-3o could yield very high GenEval scores. GenEval score is the primary metric used in this paper and highlighted in the abstract, which leads to an unfair comparison in this paper.
-  I think the benefits of integrating the real score, fake score, and generator—three models together—have been overly claimed. Particularly in the Qwen 20B example cited by the authors, all models use LoRA. For the distillation by reverse KL method, integrating two LoRAs onto the same model backbone does not bring about notable memory costs. And the GAN loss is an additional benefit; we can directly remove it and still achieve high performance.
- The combination of reverse KL and consistency-like RCGM is straightforward. Besides, similar combinations have been explored in sCM [a].
- The "twin trajectory" has been explored in FACM [b]. However, no discussion is found in the paper.

[a] Simplifying, Stabilizing and Scaling Continuous-Time Consistency Models.
[b] Flow-Anchored Consistency Models.

**Questions:**

See Weaknesses.

---

> ### Author Response · Authors · 2025-11-26
> **Reply window of W1**
>
> > Theoretical flaws. If I understand correctly, the paper trains a model with a single-timestep condition to handle generation at arbitrary steps. **However, such a model is no longer a score model, and using it in reverse KL to compute the gradient of logp is theoretically flawed**.
>
> We thank the reviewer for this comment. There may be a misunderstanding regarding the training objective, and we clarify that our trained model is not only a one-step generator, but also a score model.
>
> Our model is optimized by Eq. 10, which contains two objectives: 1) **$L_{base}$ learns multi-step and few-step, making it a score model, 2) $L_{TwinFlow}$ learns one-step and few-step, making it a one-step generator.** Therefore, the model remains a valid score estimator, and computing the gradient of $\log p$ is theoretically sound. In fact, **incorporating fake score, real score,  and one-step generator into one model is a key contribution of our work.**

---

> ### Author Response · Authors · 2025-11-26
> **Reply window of W2**
>
> > BLIP-3o-60K contains samples carefully generated using GenEval prompts, and training on BLIP-3o could yield very high GenEval scores. GenEval score is the primary metric used in this paper and highlighted in the abstract, which leads to an unfair comparison in this paper.
>
> We thank the reviewer for pointing out the potential test set overlap issue in BLIP3o dataset. BLIP3o dataset does contain GenEval-like prompts, such as *a photo of `object`*, but it does not include the same prompts in GenEval benchmarks.
>
> To eliminate the potential influence of BLIP3o data on the GenEval scores, **we conducted experiments using only the self-distilled data from Qwen-Image** (See results below)**.**
>
> Specifically, we randomly sampled approximately 100K prompts (approximately 50% of the data volume in the submitted version) from FLUX-Reason-6M [1]. We checked them using Qwen2.5-7B as a filter to ensure there does not contain any GenEval-like prompts. We synthesized corresponding images using Qwen-Image (with cfg=4.0, inference steps=50).
>
> Employing the same training configuration as in the submitted version, we obtained the following results. These results show that excluding BLIP3o data leads to only a slight drop (approximately 1%) in GenEval score, yet using self-distillation data alone still achieves high performance on GenEval, DPG-Bench, and WISE.
>
> |                                           | **NFE** | **GenEval** | **DPG-Bench** | **WISE** |
> | ----------------------------------------- | ------- | ----------- | ------------- | -------- |
> | **Containing BLIP3o (submitted version)** | 1       | 0.86        | 86.52         | 0.54     |
> | **Purely self-distilled data**            | 1       | 0.85        | 84.92         | 0.51     |
> | **Containing BLIP3o (submitted version)** | 2       | 0.87        | 87.64         | 0.57     |
> | **Purely self-distilled data**            | 2       | 0.86        | 85.30         | 0.54     |
>
> [1] [https://huggingface.co/datasets/LucasFang/FLUX-Reason-6M](https://huggingface.co/datasets/LucasFang/FLUX-Reason-6M)

---

> ### Author Response · Authors · 2025-11-26
> **Reply window of W3**
>
> > I think the benefits of integrating the real score, fake score, and generator—three models together—have been overly claimed. Particularly in the Qwen 20B example cited by the authors, all models use LoRA. For the distillation by reverse KL method, integrating two LoRAs onto the same model backbone does not bring about notable memory costs. And the GAN loss is an additional benefit; we can directly remove it and still achieve high performance.
>
> We thank the reviewer for this insightful comment. We clarify that:
>
> - **Under pure LoRA case:** the real score is the fixed teacher (20B), the generator, , fake score can be implemented as two LoRA (r=64), each of 420M trainable parameters.
> - **Under Full parameter training case:** the real score is the fixed teacher (20B), the generator is a trainable model (20B).
>   - If we implement fake score as a individual trainable model (20B), which will significantly increase GPU memory demand. After our test, this causes OOM although we use FSDPv2 **(20B fixed, 40B trainable).**
>   - Another choice is to implement fake score as LoRA (r=64), which adds 420M trainable parameters **(20B fixed, 20B+420M trainable).**
> - **Under Full parameter training case (ours):** the three models are all in one, there is only 20B parameters for all, reducing GPU memory overhead by more than half **(20B trainable).**
>
> To further explain, we add full parameter training on Qwen-Image-20B and compared with methods such as VSD, SiD, and DMD, which all have 3 models at the same time: generator, real score, and fake score.
>
> As we analyzed, under full-parameter training with a 20B model, maintaining the generator (20B), real score (20B), and fake score (20B) as separate models incurs substantial GPU memory overhead, which directly leads to OOM. To address this, we make a trade-off: the generator and real score remain as independent models, while the fake score is implemented using LoRA. **Even so,** **the framework can only run with a relatively small batch size when using FSDPv2.** If we integrate all three models into a single model, the memory demand would be reduced by more than half, and the implementation would become much simpler.
>
> The results show that our method is simple to scale on large models and can achieve high performance. The detailed comparison results are also shown in Table 9 in the revised PDF.
>
> | **Full Param Training (20B)** | **NFE** | **GenEval** | **DPG-Bench** | **WISE** |
> | ----------------------------- | ------- | ----------- | ------------- | -------- |
> | **VSD (raw)**                 | N/A     | OOM         | OOM           | OOM      |
> | **DMD (raw)**                 | N/A     | OOM         | OOM           | OOM      |
> | **SiD (raw)**                 | N/A     | OOM         | OOM           | OOM      |
> | **VSD**                       | 1       | 0.67        | 84.44         | 0.22     |
> | **VSD**                       | 2       | 0.73        | 86.16         | 0.34     |
> | **DMD**                       | 1       | 0.81        | 84.31         | 0.47     |
> | **DMD**                       | 2       | 0.80        | 84.08         | 0.46     |
> | **SiD**                       | 1       | 0.77        | 87.05         | 0.42     |
> | **SiD**                       | 2       | 0.78        | 86.94         | 0.41     |
> | **Ours**                      | 1       | 0.85        | 85.44         | 0.51     |
> | **Ours**                      | 2       | 0.86        | 86.35         | 0.55     |
> | **Ours (longer training)**    | 1       | **0.89**    | **87.54**     | **0.57** |
> | **Ours (longer training)**    | 2       | **0.90**    | **87.80**     | **0.59** |

---

> ### Author Response · Authors · 2025-11-26
> **Reply window of W4**
>
> > The combination of reverse KL and consistency-like RCGM is straightforward. Besides, similar combinations have been explored in sCM [a].
>
> We thank the reviewer for this comment. We respectfully point out that our core contribution is not the combination of reverse KL and consistency-like objectives, but rather the novel **"one model" design**. To clarify:
>
> - **Regarding sCM [a]:** The primary contribution of sCM lies in addressing the continuous limit of Consistency Models (where $\Delta t \to 0$, c.f. Algorithm 3) and utilizing Jacobian-Vector Products (JVP) to handle this regime. It does not explore the "one model" architecture we propose.
> - **Regarding our contribution:** Our method focuses on unifying the generator, real score, and fake score into one model, while requiring no standard adversarial training. This design choice addresses the high memory demand and training complexity of the previous few-step methods on large models, which was not the focus of sCM.
>
> We are encouraged that **Reviewer qqUP recognized the significance of this design**, stating that: “I like its feature where only one model is trained, and that it achieves good performance on T2I models without GAN. Compared to SiD and VSD, it combines the pretrained teacher and the true score net, while makes the fake score net online, which is a good improvement.”

---

> ### Author Response · Authors · 2025-11-26
> **Reply window of W5**
>
> > The "twin trajectory" has been explored in FACM [b]. However, no discussion is found in the paper
>
> We thank the reviewer for this insightful comment. We clarify that “twin trajectory” is not similar to FACM, the **purposes and mechanisms of FACM [b] and our TwinFlow are fundamentally different.**
>
> 1. **Difference in Purpose**
>
> - FACM: The primary purpose of FACM's approach is **stabilization**. It is explicitly designed to stabilize the unstable continuous-time Consistency Model (CM) objective by utilizing a mixed objective (part CM shortcut loss, part standard flow matching loss).
> - Our TwinFlow: The purpose of our TwinFlow is **rectification via a self-adversarial framework**. Our method is not designed to stabilize a CM objective. Instead, it creates a discriminator-free, self-adversarial process where the model learns to distinguish its own "fake" trajectories from "real" trajectories, thereby rectifying the velocity field.
>
> 1. **Difference in Mechanism**
>
> - FACM: FACM's mechanism is **separation**. Its "Expanded Time Interval" (as shown in their Figure 2.A) is an implementation strategy. It provides separate conditioning signals to manage two different tasks (the CM loss and the flow matching loss).
> - Our TwinFlow: Our mechanism is **comparison**. The "Twin Trajectory" (on $t \in [-1, 1]$) does not separate two different tasks. Instead, it creates two velocity fields that are compared against each other:
>   • real trajectory ($t \in [0, 1]$): A standard path from noise to real data.
>   • fake trajectory ($t \in [-1, 0]$): A self-adversarial path from noise to fake data.
>
> In short, our flow matching objective on $t \in [-1, 0]$ is designed to generate "fake" samples for comparison, which is a fundamental conceptual difference from using a time interval to stabilize a CM objective in FACM.

---

> ### Author Response · Authors · 2025-11-28
> **Kind Reminder for Reviewer xzEE**
>
> Dear Review xzEE,
>
> We greatly appreciate your feedback and review of our paper. Your reviews have been crucial in helping us refine and strengthen our work. In response to the concerns you raised, **we have provided detailed explanations and added new experiments to address each point.** Below, we summarize our key responses to your feedback:
>
> - **Theoretical validity:** We clarified that our model is jointly optimized as both a score estimator and a one-step generator via $L_{base}$ and $L_{TwinFlow}$. This ensures the model remains a valid score model, making the computation of the log-probability gradient in reverse KL theoretically sound.
> - **New experiments on self-distilled data:** To address concerns regarding potential data overlap in BLIP-3o, we conducted additional experiments using purely self-distilled data (filtered to exclude GenEval prompts). The results show only a marginal performance difference (~1%), confirming that our performance stems from the method itself rather than data leakage.
> - **Validation of "one model" benefits:** We provided new full-parameter training comparisons on a Qwen-Image-20B. While baseline methods (VSD, SiD, DMD) faced OOM or required complex memory trade-offs, our "one model" design significantly reduced memory overhead by over 50% while achieving superior results (GenEval 0.90).
> - **Novelty and distinction:** We distinguished our work from sCM and FACM, clarifying that our core contribution is the unified "one model" design for efficiency. We further explained that our "Twin Trajectory" is designed for self-adversarial rectification, which is fundamentally different from the stabilization mechanism used in FACM.
>
> We greatly appreciate the time and effort you've dedicated to reviewing our paper. Your feedback has been invaluable in improving the quality of our manuscript.
>
> We believe we have addressed all of your concerns and kindly ask that you reconsider your rating. If you have any additional questions or comments, please feel free to reach out.

---

> ### Comment · Reviewer_xzEE · 2025-11-28
>
> Thanks for the response. Some of my concerns have been resolved. However, I noticed that after using self-distilled data, TwinFlow's performance degraded significantly, falling short of Qwen-Image-Lightning's results, which notably weakens the promising extent of the main results in the paper. Besides, to my knowledge, Qwen-Image-Lightning's training is image-free, while TwinFlow relies on images to complete training, and its performance seems to be sensitive to dataset selection.
>
> More importantly, **my concerns about the theoretical flaws have not been addressed**: A score model refers to a model trained by score matching loss, whose optimal solution converges to the score of the data distribution. Twinflow is trained with a mixture of multiple losses including: score loss, consistency-like loss, and RKL loss, and only has one timestep as a condition. A model trained in this way is clearly no longer a score model. *We cannot claim that a model can be used to estimate score just because its training loss includes score loss. Therefore, using Twinflow to compute the score for applying RKL is theoretically flawed.*
>
> There highly likely exists some other mechanism behind why Twinflow can work, and I encourage the authors to carefully investigate this mechanism in the future. However, as it stands, this paper lacks understanding of this mechanism, and attributing the success to RKL loss may be misleading to the community.

---

> ### Author Response · Authors · 2025-12-01
> **Follow-up Response 1**
>
> Thanks for the reviewer’s response. ***We respectfully believe that the reviewer's response contains misunderstandings regarding comparative baselines and factual errors about the theoretical foundations.*** We provide detailed clarifications below.
>
> - **Our method’c core contribution is the one-model design and we are NOT follow-up work of Qwen-Image-Lightning.** Furthermore, given the lack of an official technical report or released training details for Qwen-Image-Lightning, it is a unfair comparison regarding method superiority solely through benchmark results.
> - We respectfully disagree that the performance variation using self-distilled data constitutes a "significant degradation" or implies fragility.
>     - First, using self-distilled data results in **GenEval and WISE score stability** and a DPG-Bench drop of 2.87%. **This confirms our method works robustly without relying on specific ‘golden’ datasets** (like BLIP3o) to achieve high performance. **We believe this addresses the reviewer’s initial concern on GenEval score (W2).**
>     - Second, the reviewer compares our self-distilled result (GenEval: 0.85, DPG 84.92, WISE: 0.51) to Qwen-Image-Lightning (GenEval: 0.85, DPG 87.79, WISE: 0.51) and concludes we are falling short. We would like to point out:
>         - Point-to-point benchmark comparison is flawed because **Qwen-Image-Lightning suffers from severe mode collapse (low diversity)**. As noted in our response to Reviewer ESMU (Reply to Q8, also see Fig. 5 in the paper), **Qwen-Image-Lightning yields a low LPIPS diversity score (higher is better) (\~0.30) compared to TwinFlow (\~0.50).**
>         - DPG-Bench evaluates 4 images generated by the model for each prompt (different noises), Qwen-Image-Lightning **is thus gaming DPG-Bench by generating nearly identical images, which makes the evaluation easier. This exploitation helps Qwen-Image-Lightning achieves slightly higher score.**
>         - Thus, only one flawed benchmark result (DPG-Bench) is slightly lower does not effectively support ‘falling short’.
> - Due to Qwen-Image-Lightning has no official public technical report or released training details, we cannot identify the reviewer’s claim of image-free training. Assuming that image-free holds, we proceed with the following analysis:
>     - Image-free means only using prompts. Such method typically needs the model itself to do one forward to get a predicted  $\hat{x}$  as the ‘image’.
>     - When using self-distilled data, we are also using the Qwen-Image itself to predict images (with many forwards), this can already be viewed as a type of weak image-free.
>     - As the experiments showed and we have clarified before, the results of using only self-distilled data show stability across the benchmarks.
>     - **Finally we would like to clarify that our method does not focus on image-free, our core contribution lies in one-model design.**
> - **Conclusion:** **Our model maintains high diversity and competitive scores, proving the method's efficacy is not dependent on specific dataset selection.**

---

> ### Author Response · Authors · 2025-12-01
> **Follow-up Response 2**
>
> - We believe the reviewer has factual misunderstandings regarding our model's input space and training objective.
>     - ***First, the statement that our method “only has one timestep as a condition” is a significant factual error.***
>         - **TwinFlow does not only condition on a single time step; it explicitly uses two timesteps as conditions.** As defined in Eq. 1, the base loss $L_{base}$ in TwinFlow is an *any-step* loss, **conditioned on both the current time $t$ and the target time $r$** (see L147–154 in the paper), where $t,r \sim \mathrm{U}(0, 1), r<t$. $L_{base}$ learns any $t$ to $r$ transitions on PF-ODE (c.f. Eq. 6&7&8 in RCGM). Also, in MeanFlow, Eq. 4-9 also derive how the model learns any-step transitions. In Sec. 2 of our revised PDF (L170-L71), we also introduced that MeanFlow is a $N=1$ formulation in RCGM.
>         - **Specifically, when $r \to t$, the model estimates $\frac{d x_{t}}{d t}$, which is the instant velocity field,** i.e. the multi-step objective. Under Gaussian Probability Path (such as OT Path), velocity field and score is linearly related. Learning velocity field is equivalent to learning score. **Thus, $L_{base}$ effectively enables the model to learn a score function.**
>         - Since the model is a score model, we can correctly compute the gradient of $\log p$. Thus, the assertion that “A model trained in this way is clearly no longer a score model. *We cannot claim that a model can be used to estimate score just because its training loss includes score loss. Therefore, using Twinflow to compute the score for applying RKL is theoretically flawed.*” is not valid.
>     - Second, to avoid confusion, we have added a sentence in the revised PDF explicitly stating that the implementation requires conditioning on two timesteps (L157-158).
>     - Additionally, to aid understanding, we provide a minimal example network implementation on the Moons dataset:
>
> ```python
> class MLP(nn.Module):
>     def __init__(self, in_dim, context_dim, h, out_dim):
>         super(MLP, self).__init__()
>         # generator
>         self.network = nn.Sequential(
>             nn.Linear(in_dim + context_dim, h),
>             nn.ReLU(),
>             nn.Linear(h, h),
>             nn.ReLU(),
>             nn.Linear(h, out_dim),
>         )
>
>     def forward(self, x_t, t, r):
>         # x_t: noisy data
>         # t: current time
>         # r: target time
>         t = t.flatten().unsqueeze(1)
>         r = r.flatten().unsqueeze(1)
>         input = torch.cat((x_t, t, r), dim=1)
>         return self.network(input)
> ```
>
> Finally, **we again would like to point out that TwinFlow’s theory foundation is clear and sound.**
>
> - First, as Eq.1 shows, $L_{base}$ already ensures the model learns a valid score function. i.e. when $r \to t$, the model estimates $\frac{d x_{t}}{d t}$. **$L_{base}$ can learn the real score.**
> - Second, the first part of $L_{TwinFlow}$, i.e. $L_{adv}$ is a standard flow matching loss, but its $t\in [-1,0]$. **$L_{adv}$ aims to learn fake score on the negative time interval.**
> - Third, the second part of $L_{TwinFlow}$, i.e. $L_{rectify}$ is a distribution matching-like loss. Given that $L_{base}$ has already effectively learned the score function, we derive a distribution-matching objective via the reverse KL, which ultimately leads to a velocity-matching formulation and yields a tractable loss. **$L_{rectify}$ aims to learn the generator.**
> - Through $L_{base}$ and $L_{TwinFlow}$, we complete our one-model design, which unifies generator, fake score, and real score into one model.
>
> We would like to point out again that **our core contribution lies in the “one-model” design**, which unifies the one-step generator, fake score, and real score into a single model—enabling efficient and scalable training on large models. To achieve, we combine $L_{base}$ and $L_{TwinFlow}$.
>
> As we have illustrated in the paper and the ablation studies (Reply to reviewer ESMU and qqUP) added in the rebuttal. We believe the mechanism of $L_{base}$ and  $L_{TwinFlow}$ is clearly demonstrated.

---

### Author Response · Authors · 2025-11-26
**General Response**

We sincerely thank all reviewers for their insightful feedback, which has greatly helped us refine the manuscript.

Key updates in the rebuttal:

- **Manuscript revisions:** We have **clarified the notations in Section 2 & 3 and corrected minor typos** throughout the text. All changes are highlighted in green for ease of review. We also add more visualizations in the appendix (Fig. 12-15).
- **Full-parameter training on Qwen-Image-20B:** We have extended TwinFlow to support full-parameter training on the Qwen-Image-20B model. Results are reported in **Table 9 of the revised manuscript.**
- **Detailed baseline comparison:** We provide a detailed comparison of our method against VSD, SiD, and DMD under full-parameter training on Qwen-Image-20B. These results are also included in **Table 9 of the revised manuscript.**
  - **Implementation details**
    - **Data:** Since DMD employs a regression loss, we need to pre-generate offline data using Qwen-Image-20B. To ensure a fair comparison with established baselines, we randomly sample 50K prompts from the datasets we used, and generate images using Qwen-Image-20B. All experiments are conducted on this self-generated dataset.
    - **Model:** Qwen-Image-20B
      - The generator is implemented by a **fully trainable Qwen-Image-20B**
      - The real score is implemented by a fixed Qwen-Image-20B teacher model
      - The fake score is implemented by a LoRA (rank=64) due to inadequate memory
      - **Our method: simply one trainable 20B model**
  - **Other training configs:** We employ FSDPv2 for full-parameter training. The effective global batch size is 32 with a local batch size of 4 per device.
- **No-GenEval experiments**: we use pure self-distilled data (without GenEval-like prompts) and still achieve high performance on all benchmarks.
- **Capacity** **experiments:** we ablate LoRA ranks of 32, 64, and 128, and the results demonstrate that our **one-model design does not introduce parameter capacity limitations and achieves strong performance across all benchmarks.**

---

### Meta-Review · Area_Chair_Mmhz · 2025-12-19

**Summary:**

This paper focuses accelerating image generation model by reducing number of function evaluations. The proposed method provides comparable performance while accelerating the pre-trained model by 100 times. In the inital review, reviewers rate this paper with score 4,6, and 6. The concerns mainly lie in therotical flaws (Reviewer xzEE), unfair and lack of comparison (all reviewers), LoRA training (Reviewer xzEE and Reviewer ESMU) and unclear writing (Reviewer ESMU and Reviewer qqUP). During rebuttal, authors provide comprehensive additional experiment and analysis on their method and revise the paper to make it clearer.

After rebuttal, all the experimental concerns are addressed (all reviewers) and only the one main concern remains: the theoritical defination disagreement on score models by Reviewer xzEE.  Overall, AC confirms that the experiment is extensive, and the results are impressive. Although the paper received divergent scores, particularly the rejection score by xzEE, AC confirms that the merits in this paper outweigh the theoritical defination disagreement by Reviewer xzEE. AC recommends this paper to **Accept (Poster)**.

**Reviewer Concerns:**

The concerns mainly lie in therotical flaws (Reviewer xzEE), unfair and lack of comparison (all reviewers), LoRA training (Reviewer xzEE and Reviewer ESMU) and unclear writing (Reviewer ESMU and Reviewer qqUP).

The **unfair and lack of comparison** and **LoRA training** are addressed. During rebuttal, the authors provide experiments on replace the BLIP3 dataset, full model training, training with different LoRA configuration, and loss ablation. Also, they provide additional comparison with previous method on efficiency, effectiveness and computational cost.

The **unclear writing** is addressed. The authors revise the paper and provide additional explanation during rebuttal.

The **therotical flaws concern remains unsolved**. The definition of score models disagrees between Reviewer xzEE and the authors. AC confirms that the merits in this paper outweigh the theoritical defination disagreement.

**Reviewer Scores:**

**Reviewer xzEE**: keep 4
Reviewer xzEE participates in the discussion. Based on his comment, he disagrees with the author on this most important concern.

**Reviewer ESMU**: keep 6 or raise to 8.
Reviewer ESMU does not participates in the discussion, but his concerns and questions are all responded by the authors. AC thinks his concerns are mostly addressed.

**Reviewer qqUP**: keep 6
Reviewer qqUP participates in the discussed and decided to keep his rating.

---

### Decision · Program_Chairs · 2026-01-26

Accept (Poster)